# Risk-controlling Prediction with Distributionally Robust Optimization

**Franck Iutzeler**  *franck.iutzeler@math.univ-toulouse.fr*
*Institut de Mathématiques de Toulouse*
*Université de Toulouse, CNRS, UPS, 31062, Toulouse, France*

**Adrien Mazoyer**  *adrien.mazoyer@math.univ-toulouse.fr*
*Institut de Mathématiques de Toulouse*
*Université de Toulouse, CNRS, UPS, 31062, Toulouse, France*

**Reviewed on OpenReview:** *https://openreview.net/forum?id=d9dl6DyJpJ*

## Abstract

Conformal prediction is a popular paradigm to quantify the uncertainty of a model's output on a new batch of data. Quite differently, distributionally robust optimization aims at training a model that is robust to uncertainties in the distribution of the training data. In this paper, we examine the links between the two approaches. In particular, we show that we can learn conformal prediction intervals by distributionally robust optimization on a well chosen objective. This further entails to train a model and build conformal prediction intervals all at once, using the same data.

## 1 Introduction

### 1.1 Background and contribution

During the last decade, the use of conformal inference to quantify reliability of black-box method and complex machine learning models, has been widely studied (see e.g., (Vovk et al., 2005; Angelopoulos and Bates, 2023; Fontana et al., 2023)). In particular, Split Conformal Inference (SCI) as introduced by Papadopoulos et al. (2002) is a procedure that aims to estimate the reliability of pre-trained models by observing its performance on *calibration* data. More formally, for some input/output spaces $\mathcal{X} \times \mathcal{Y}$, the aim is to construct a suitable function $C : \mathcal{X} \to \mathsf{P}(\mathcal{Y})$, with output in the set of parts of $\mathcal{Y}$ denoted by $\mathsf{P}(\mathcal{Y})$, from the model $f : \mathcal{X} \to \mathcal{Y}$ and $n$ calibration points $(X_i, Y_i)_{i=1}^n$ in $\mathcal{X} \times \mathcal{Y}$. In SCI, the objective is that for a new point $(X_{n+1}, Y_{n+1})$ and a miscoverage level $\alpha$ set by the user, the following holds

$$\mathbb{P}[Y_{n+1} \in C(X_{n+1})] \geq 1 - \alpha, \tag{1}$$

where the above probability is taken with respect to the joint distribution of $(X_i, Y_i)_{i=1}^{n+1}$. The main advantage of SCI is that it requires only exchangeability assumptions for the joint distribution of $(X_i, Y_i)_{i=1}^{n+1}$ (see the references above for more details), and still gives finite sample guarantees.

We are considering here a related paradigm to SCI, referred to as *risk controlling prediction* by Bates et al. (2021). The controlled risk corresponds to $\mathcal{R}(\mathcal{T}_\lambda) := \mathbb{E}[L(Y_{n+1}, \mathcal{T}_\lambda(X_{n+1}))]$, where $\mathcal{T}_\lambda : \mathcal{X} \to \mathsf{P}(\mathcal{Y})$ is a parameterized set-valued predictor and $L : \mathcal{Y} \times \mathsf{P}(\mathcal{Y}) \to \mathbb{R}_+$ is a loss that evaluates the quality of $\mathcal{T}_\lambda$. The objective here is to ensure that, given some $\hat{\lambda}_n$ depending on the calibration data, the risk $\mathcal{R}(\mathcal{T}_{\hat{\lambda}_n}) := \mathbb{E}[L(Y_{n+1}, \mathcal{T}_{\hat{\lambda}_n}(X_{n+1}))|(X_i, Y_i)_{i=1}^n]$ remains with high probability below some target level. More formally we aim to guarantee for levels $\alpha$ and $\delta$ chosen by the user that

$$\mathbb{P}\left[\mathcal{R}(\mathcal{T}_{\hat{\lambda}_n}) \leq \alpha\right] \geq 1 - \delta, \tag{2}$$

where the above probability is over the joint distribution of $(X_i, Y_i)_{i=1}^n$. The only requirement to get that guarantee is assuming the calibration points $(X_i, Y_i)_{i=1}^n$ to be independent and identically distributed (i.i.d.) and having access to a concentration inequality in order to bound $\mathcal{R}(\mathcal{T}_\lambda)$ by some $\alpha$ with probability $1 - \delta$.

We develop in this paper the idea that an interesting alternative to concentration inequalities in this context can be obtain by the formalism of distributionally robust optimization (DRO). Indeed, the principle of distributional robustness is to design an uncertainty neighborhood of distributions $\mathcal{U}(\mathrm{P_n})$ around the empirical distribution of the calibration data $\mathrm{P_n} = \frac{1}{n} \sum_{i=1}^n \delta_{(X_i, Y_i)}$. Then, if the distribution of $(X_{n+1}, Y_{n+1})$ belongs to $\mathcal{U}(\mathrm{P_n})$ with probability $1 - \delta$, then (2) holds with $\alpha = \alpha(\hat{\lambda}_n)$ where

$$\alpha(\lambda) = \sup_{\mathrm{Q} \in \mathcal{U}(\mathrm{P_n})} \mathbb{E}_{(X,Y) \sim \mathrm{Q}}[L(Y, \mathcal{T}_\lambda(X))] \tag{3}$$

is a distributionally robust version of the conformal loss. Conversely, from the user-chosen risk level $\alpha$, choosing $\hat{\lambda}_n$ so that $\alpha(\hat{\lambda}_n) \leq \alpha$ leads to the sought guarantee (2). The design effort for the upper-bound is thus differed to the construction of $\mathcal{U}(\mathrm{P_n})$.

Conformal methods mainly rely on exchangeability assumptions on $(X_i, Y_i)_{i=1...n+1}$ which may be violated, typically when the calibration points are i.i.d. according to a distribution different from the test point. Interestingly, the distributional robustness approach presented above does not rely on such assumptions but only on the probability of the distribution of $(X_{n+1}, Y_{n+1})$ belonging to $\mathcal{U}(\mathrm{P_n})$.

In this paper, we consider the paradigm of Risk-Controlling Prediction Sets (RCPS), i.e., (2), where the upper-bound on the risk is obtained by Wasserstein distributionally robust optimization (WDRO), i.e., (3) where $\mathcal{U}(\mathrm{P_n})$ is a ball of radius $\rho$ around $\mathrm{P_n}$ in the Wasserstein distance. We demonstrate that the procedure exposed by Bates et al. (2021) remains valid provided that $\rho$ is adequately chosen and can additionally suffer a shift compared to the training distribution in terms of Wasserstein distance.

In Section 2, we recall the RCPS methodology and connect it to distributional robustness. Then, in Section 3, we leverage tools from Wasserstein distributionally robust optimization to show the following result (stated below informally):

**Theorem 1** (informal). *Let $(X_i, Y_i)_{i=1}^n \sim \mathrm{P}^{\otimes n}$ and $(X_{n+1}, Y_{n+1}) \sim \mathrm{P_\star}$. For any $\alpha > 0$ and any $\delta \in (0,1)$, there exists $\rho(n, \delta)$ and $\hat{\lambda}_n$ (chosen so that $\alpha(\hat{\lambda}_n) \leq \alpha$ where $\alpha(\cdot)$ is defined in (3)) so that for any $\rho \geq \rho(n, \delta)$*

$$\mathbb{P}\Big[\mathcal{R}(\mathcal{T}_{\hat{\lambda}_n}) \leq \alpha\Big] \geq 1 - \delta$$

*whenever $W_2(\mathrm{P}, \mathrm{P_\star}) \leq \rho - \rho(n, \delta)$.*

This enables theoretically the use of WDRO to produce upper-bounds in RCPS. Interestingly, this method is pertinent for any value of $\alpha$ (see Section 3.2.3) and is compatible with a joint training of the prediction and conformal prediction on the same dataset (see Section 3.5). Finally, we illustrate its encouraging performance in practice in Section 4.

## 1.2 Related works

The link between classical SCI and RCPS can be made through the work of Gupta et al. (2022) who reformulated Conformal Inference (CI) in terms of nested prediction sets, as well as Conformal Risk Control (CRC) introduced by Angelopoulos et al. (2024) where the guarantee (2) is expressed in expectation, that is $\mathbb{E}[\mathcal{R}(\mathcal{T}_{\hat{\lambda}_n})] \leq \alpha$. Note that guarantee (1) is then retrieved by taking $\mathcal{R}(\mathcal{T}_{\hat{\lambda}_n}) = \mathbb{1}_{Y_{n+1} \notin \mathcal{T}_{\hat{\lambda}_n}(X_{n+1})}$, where the expression of $\hat{\lambda}_n$ depends on the calibration data (see Bates et al. (2021, App. A) for more details).

Distribution shift in CI has been recently widely studied for different type of shifting. Tibshirani et al. (2019) studied the covariate shift case (i.e., $X$ distribution differs betwen calibration and test points, but not the $Y|X$ distribution) by introducing the notion of weighted exchangeability. This approach has been generalized for the CRC context by Angelopoulos et al. (2024, Sec. 4.1). Podkopaev and Ramdas (2021) used similar arguments in order to tackle the label shift case (i.e., $Y$ distribution differs but not the $X|Y$ distribution). Barber et al. (2023) studied the consequences of such distribution shifts, by quantifying the

coverage gap in terms of total variation distance. The *online* time series case has also been considered in the works of Gibbs and Candès (2021; 2024). Their former work has been considered in the CRC context by Feldman et al. (2023). Cauchois et al. (2024) studied shift in terms of $f$-divergence, which a closest context of our, and less restrictive than the other case mentioned here. It is also worth to notice that our results are also close but different from the conformal prediction with adversarial shift presented by Bates et al. (2021, Sec. 6.3) as a possible extension, since we consider shifts on the data distribution rather than pointwise shift in the training set. That adversarial context has actually been studied for SCI by Gendler et al. (2022) followed by Yan et al. (2024).

Very recently, Aolaritei et al. (2025) proposed to address the problem of distribution shifts in conformal prediction by considering ambiguity sets of distributions (similar to what we denote by $\mathcal{U}(\mathrm{P_n})$) based on the Lévy–Prokhorov pseudo-metric (where we use the Wasserstein distance).[1] Their results are thus complementary to ours.

## 2 Risk-controlled prediction sets and Distributional Robustness

In this section, we focus on the RCPS paradigm. In the third part of the section, we connect this with the notion of distributional robustness.

### 2.1 Setup

We suppose to have access to a sample $(X_i, Y_i)_{i=1}^n$ of i.i.d. couples of random variables with probability distribution $\mathrm{P_0}$ on $\mathcal{X} \times \mathcal{Y}$.

We seek to provide conformal prediction sets around a predictor $f : \mathcal{X} \to \tilde{\mathcal{Y}}$ (the predictor output space $\tilde{\mathcal{Y}}$ is usually equal to $\mathcal{Y}$ but in some contexts such as classification, it can be equal to the simplex over $\mathcal{Y}$). For the moment, we will consider that the predictor is fixed (e.g., learned by an external procedure), we will get back to the predictor training in Section 3.5.

We recall here the notion of nested prediction sets as described by Bates et al. (2021). We consider a class of parametric sets of the form

$$\begin{aligned} \mathcal{T} : \quad \Lambda \times \mathcal{X} &\to \mathsf{P}(\mathcal{Y}) \\ (\lambda, x) &\mapsto \mathcal{T}_\lambda(x) \end{aligned}$$

where $\mathsf{P}(\mathcal{Y})$ denotes the parts of set $\mathcal{Y}$ and $\Lambda$ is a closed subset of $\mathbb{R}$. We assume that these predictions sets are *nested* i.e., that

$$[\lambda_1 \geq \lambda_2] \Rightarrow [\mathcal{T}_{\lambda_1}(x) \subseteq \mathcal{T}_{\lambda_2}(x) \text{ for all } x \in \mathcal{X}]. \tag{4}$$

Typically, these sets are constructed from predictor outputs, as detailed in the examples below.

Finally, we define the loss for the conformal prediction as

$$\begin{aligned} L : \quad \mathcal{Y} \times \mathsf{P}(\mathcal{Y}) &\to \mathbb{R}_+ \\ (y, S) &\mapsto L(y, S) \end{aligned}$$

which we will often use in conjunction of the parametric sets above by considering $L(y, \mathcal{T}_\lambda(x))$. Moreover, for practical convenience and in line with applications of interest, we will choose it *monotone* in its second argument i.e.,

$$[S \subset S'] \Rightarrow [L(y, S) \geq L(y, S') \text{ for all } y \in \mathcal{Y}]. \tag{5}$$

The goal in conformal prediction is to use the available samples to build prediction sets around a predictor output such that the average conformal prediction loss is the low in expectation over the unknown distribution from which the samples are generated. To do so, we rely on the following assumption.

---

[1] The authors comment on the links between the Lévy–Prokhorov pseudo-metric and the *type*-$\infty$ Wasserstein distance, which is very different to the *type*-2 Wasserstein distance we employ both in terms of statistical guarantees and computational issues.

**Assumption 2.** *The following properties hold:*

   *i) the parametric sets $\{\mathcal{T}_\lambda\}_{\lambda \in \Lambda}$ satisfy the nesting property (4)*

   *ii) the conformal loss $L$ is non-negative, upper-semicontinuous, and satisfies the monotony property (5)*

   *iii) the pointwise limit of the function $(\lambda, (x, y)) \mapsto L(y, \mathcal{T}_\lambda(x))$ as $\lambda \to \sup(\Lambda)$ is the null function*

   *iv) the function $(x, y) \mapsto L(y, \mathcal{T}_\lambda(x))$ is upper-bounded uniformly in $\lambda$.*

Assumption 2-i) is the classical nesting property of conformal prediction. Item ii) is also standard and fits the vast majority of the target usecases (the upper-semicontinuity assumption can be relaxed for most results of the paper). Note also that non-monotone losses could be also considered, using procedures described by Angelopoulos et al. (2024)[2]. As for item iii), it is slightly different from e.g., Bates et al. (2021) which assumes the existence of some $\lambda_{\max}$ such that $\mathbb{E}_{(X,Y) \sim P_\star}[L(Y, \mathcal{T}_{\lambda_{\max}}(X))] = 0$ (as well as all greater $\lambda$ by monotony of $L$); we rely on this modified assumption mainly to avoid complications about distributions supports in future developments. Notice that item iv) is actually not mandatory for RCPS: it can be relaxed into assuming the coefficient of variation of the empirical risk (6) is bounded, in order to consider unbounded losses (see Bates et al. (2021, Sec. 3.2)). However, it will be required when introducing the connection with Wasserstein distributionally robust optimization in Section 3. Moreover, without any loss of generality, we will assume that losses are bounded by 1.

**Example 2.1** (Regression). *Take $\mathcal{X} = \mathbb{R}^d$ and $\mathcal{Y} = \mathbb{R}$. Typical predictors include (pretrained) linear models ($f_\theta(x) = \langle \theta; x \rangle$), neural networks, etc. A simple class of parametric sets fitting our framework is $\mathcal{T}_\lambda(x) = \{y \in \mathcal{Y} : |y - f_\theta(x)| \leq \lambda\}$ (in this setting, $\tilde{\mathcal{Y}} = \mathcal{Y} = \mathbb{R}$). Doing so, the conformal loss can be defined from the 0/1 loss as $L(y, \mathcal{T}_\lambda(x)) = \mathbb{1}_{y \notin \mathcal{T}_\lambda(x)} = \mathbb{1}_{|y - f_\theta(x)| > \lambda}$.*

**Example 2.2** (Multi-class Classification). *Take $\mathcal{X} = \mathbb{R}^d$ and $\mathcal{Y} = \{1, 2, .., c\}$. Usual predictors include (pretrained) once again linear models, neural networks, etc. However, in this setup, in order to get meaningful prediction sets, it is usual to work with predictors outputting the probability of each class rather than a single class, so $\tilde{Y}$ is the simplex over $c$ alternatives (This is not really restrictive since this is exactly the output of a linear predictor or a neural net after the cross-entropy but before thresholding). A possible class of parametric sets is $\mathcal{T}_\lambda(x) = \{y \in \mathcal{Y} : f_\theta(x)[y] \geq 1 - \lambda\}$ (most probable classes predicted, in the sense of a probability greater than the threshold), associated to a loss of the form $L(y, \mathcal{T}_\lambda(x)) = \mathbb{1}_{y \notin \mathcal{T}_\lambda(x)} = \mathbb{1}_{f_\theta(x)[y] < 1 - \lambda}$.*

### 2.2 Risk-controlled prediction sets

The objective of risk-controlled prediction as described by Bates et al. (2021) is to find some $\lambda$ that gives a low loss for a new point $(X, Y)$ distributed according to some distribution $P_\star$. More precisely, we want to ensure that

$$\mathcal{R}(\lambda) \coloneqq \mathbb{E}_{(X,Y) \sim P_\star}[L(Y, \mathcal{T}_\lambda(X))]$$

is small. Note that the expectation is taken over the distribution of the new sample, $P_\star$, which is *a priori* unknown. Importantly, $\mathcal{T}_\lambda$ (and thus $(x, y) \mapsto L(y, \mathcal{T}_\lambda(x))$) does not depend on $(X_i, Y_i)_{i=1}^n$, but does depend on $\lambda$ which is a fixed parameter. However, in what follows, the choice of $\lambda$ may depend on $(X_i, Y_i)_{i=1}^n$, in which case it will be denoted by $\hat{\lambda}_n$ to avoid confusion.

As a proxy, the empirical risk obtained from the data is an entry point for computations:

$$\widehat{\mathcal{R}}_n(\lambda) \coloneqq \mathbb{E}_{(X,Y) \sim P_n}[L(Y, \mathcal{T}_\lambda(X))] = \frac{1}{n} \sum_{i=1}^n L(Y_i, \mathcal{T}_\lambda(X_i)) \tag{6}$$

where $P_n = \frac{1}{n} \sum_{i=1}^n \delta_{(X_i, Y_i)}$ denotes the empirical distribution of the calibration data.

Whenever the samples $(X_i, Y_i)_{i=1}^n$ are i.i.d. and, even more importantly $P_0 = P_\star$, the following rationale has been proposed by Bates et al. (2021):

---

[2]At a cost though: this might lead to larger prediction set as the gap with monotonicity of the loss increases.

**Step A** Given a confidence $1 - \delta \in (0, 1)$, consider a (random) function $\widehat{\mathcal{R}}_n^+(\lambda)$ verifying

$$\forall \lambda \in \Lambda, \quad \mathbb{P}_{(X_i, Y_i)_{i=1}^n \sim \mathrm{P}^{\otimes n}} \left[ \boldsymbol{\mathcal{R}}(\lambda) \le \widehat{\mathcal{R}}_n^+(\lambda) \right] \ge 1 - \delta; \tag{7}$$

this is typically obtained by a concentration inequality (e.g., Hoeffding).

**Step B** Given a level $\alpha \in (0, 1)$, find a random value $\hat{\lambda}_n$ (depending on $(X_i, Y_i)_{i=1}^n$) such that

$$\forall \lambda \in \Lambda, \quad \lambda \ge \hat{\lambda}_n \Rightarrow \widehat{\mathcal{R}}_n^+(\lambda) \le \alpha \tag{8}$$

by using a 1D search.

Putting it all together, we obtain that

$$\mathbb{P}_{(X_i, Y_i)_{i=1}^n \sim \mathrm{P}^{\otimes n}} \left[ \boldsymbol{\mathcal{R}}(\hat{\lambda}_n) \le \alpha \right] \ge 1 - \delta \tag{9}$$

which, in words, means that we have a low true risk with high probability (this last probability reflecting the fact that $\hat{\lambda}_n$ depends on $(X_i, Y_i)_{i=1}^n$ and is thus random).

**Example 2.3** (Regression (cont.)). *With the notation from Example 2.1 and relying on the fact that* $\mathrm{P}_0 = \mathrm{P}_\star = \mathrm{P}$, *Equation (9) guarantees that*

$$\mathbb{P}_{(X_i, Y_i)_{i=1}^n \sim \mathrm{P}^{\otimes n}} \left[ \mathbb{P}_{(X, Y) \sim \mathrm{P}} \left[ |Y - f_\theta(X)| < \hat{\lambda}_n \right] > 1 - \alpha \right] \ge 1 - \delta$$

*where we recover our double guarantee on the conformal interval: with respect to the randomness of the new point and of the data sample.*

## 2.3 Distributionally Robust Optimisation

The empirical risk (6) can lead to over-confident decisions (lower risk for training samples than in test) and be sensitive to distribution shifts between training and application (see e.g., (Esfahani and Kuhn, 2018)). In our context, this means that the empirical risk may not be a valid upper bound of the true risk in the sense of (7) (hence the need of an auxiliary function $\widehat{\mathcal{R}}_n^+(\lambda)$). To overcome these drawbacks, an approach gaining momentum in machine learning is *distributionally robust* optimization, which consists in minimizing the *worst expectation* of the loss when the distribution lives in a neighborhood of $\mathrm{P}_n$.

In our context, distributionally robust optimization amount to solving the problem

$$\sup_{\mathrm{Q} \in \mathcal{U}_\rho(\mathrm{P}_n)} \mathbb{E}_{(X, Y) \sim \mathrm{Q}}[L(Y, \mathcal{T}_\lambda(X))] \tag{10}$$

where the supremum is taken over all probability distributions on $\Xi := \mathcal{X} \times \mathcal{Y}$ which are *close* to $\mathrm{P}_n$ i.e.,

$$\mathcal{U}_\rho(\mathrm{P}_n) := \{ \mathrm{Q} \in \mathcal{P}(\Xi) : \mathrm{dist}(\mathrm{P}_n, \mathrm{Q}) \le \rho \},$$

where $\rho > 0$ controls the required level of robustness around $\mathrm{P}_n$ and dist is denotes any kind of metric on probability distributions.

Popular choices of distribution neighborhoods are based on the Kullback-Leibler (KL) divergence (Laguel et al., 2020; Levy et al., 2020), kernel tools (Zhu et al., 2021a; Staib and Jegelka, 2019; Zhu et al., 2021b), moments (Delage and Ye, 2010; Goh and Sim, 2010), or the Wasserstein distance (Shafieezadeh Abadeh et al., 2015; Esfahani and Kuhn, 2018).

An important motivation for distributionally robust models is that if the true distribution belongs to the robustness neighborhood, the method directly benefits from generalization guarantees. This is why a careful choice of the neighborhood is necessary. Indeed, if $\mathrm{P}_\star \in \mathcal{U}_\rho(\mathrm{P}_n)$ then we directly get that

$$\boldsymbol{\mathcal{R}}(\lambda) := \mathbb{E}_{(X, Y) \sim \mathrm{P}_\star}[L(Y, \mathcal{T}_\lambda(X))] \le \sup_{\mathrm{Q} \in \mathcal{U}_\rho(\mathrm{P}_n)} \mathbb{E}_{(X, Y) \sim \mathrm{Q}}[L(Y, \mathcal{T}_\lambda(X))]. \tag{11}$$

This attractive property is also highly connected with the choice of the metric on probability distributions. For instance, using the KL divergence (i.e., taking $\mathcal{U}_\rho(\mathrm{P_n}) = \{\mathrm{Q} \in \mathcal{P}(\Xi) : \mathrm{KL}(\mathrm{P_n}, \mathrm{Q}) \le \rho\}$) was studied in details in the literature as a natural choice leading to tractable formulations, see e.g., Ben-Tal et al. (2013); Hu and Hong (2013); Namkoong and Duchi (2016) and Wang et al. (2023, Rem. 5). However, it imposes that the support of $\mathrm{P_\star}$ should comprised in the one of $\mathrm{P_n}$ if one wants to have $\mathrm{P_\star} \in \mathcal{U}_\rho(\mathrm{P_n})$ since the KL divergence between two distribution is infinite as soon as the supports are disjoint. Thus, this approach amounts to reweighting the samples with higher conformal loss so that the newly obtained empirical conformal loss is higher and can act as an upper-bound. While this approach does not grants us the same guarantees, we note that the idea of reweighting samples to account for distributional shifts provides a nice connection with the notion of weighted conformal prediction for covariate shifts, see Tibshirani et al. (2019) and Cauchois et al. (2024) in light of Hu and Hong (2013, Sec. 2).

In this paper, we focus on the ambiguity sets relying the Wasserstein distance, which naturally metrizes the convergence of measures,[3] which is a sought feature in this context. This is the topic of the next section.

## 3 Risk-controlled prediction sets using Wasserstein Distributionally Robust Optimization

In this section, we propose to use a Wasserstein distributionally robust version of the empirical risk as our high probability upper-bound on the true risk in the form of (7). We investigate the tractability and the statistical guarantees that can be offered by WDRO RCPS.

We recall that:

- we denote by $\mathrm{P_\star}$ the unknown true distribution of $(X, Y)$, so that the true risk is $\mathcal{R}(\lambda) := \mathbb{E}_{(X,Y)\sim\mathrm{P_\star}}[L(Y, \mathcal{T}_\lambda(X))]$

- we have access to $n$ samples $(X_i, Y_i)_{i=1}^n$, i.i.d. with probability distribution $\mathrm{P_0}$ (different from $\mathrm{P_\star}$ in general)

- we denote by $\mathrm{P_n} = \frac{1}{n}\sum_{i=1}^n \delta_{(X_i,Y_i)}$ the empirical data distribution, so that the empirical risk is $\widehat{\mathcal{R}}_n(\lambda) := \mathbb{E}_{(X,Y)\sim\mathrm{P_n}}[L(Y, \mathcal{T}_\lambda(X))] = \frac{1}{n}\sum_{i=1}^n L(Y_i, \mathcal{T}_\lambda(X_i))$

### 3.1 Wasserstein distributionally robust optimization

Wasserstein distributionally robust optimization corresponds to Problem (10) with distribution neighborhoods formulated in terms of optimal transport distance i.e.,

$$\mathcal{U}_\rho(\mathrm{P_n}) := \{\mathrm{Q} \in \mathcal{P}(\Xi) : W_2(\mathrm{P_n}, \mathrm{Q}) \le \rho\},$$

where $W_2(\cdot, \cdot)$ denotes the (type-2) Wasserstein distance[4] defined as

$$W_2(\mathrm{Q}, \mathrm{Q}') := \left(\inf_{\pi\in\mathcal{P}(\Xi\times\Xi),\pi_1=\mathrm{Q},\pi_2=\mathrm{Q}'} \mathbb{E}_{(\xi,\zeta)\sim\pi}\left[\|\xi-\zeta\|^2\right]\right)^{1/2},$$

where $\mathcal{P}(\Xi \times \Xi)$ is the set of probability distributions in the product space $\Xi \times \Xi$, and $\pi_1$ (resp. $\pi_2$) denotes the first (resp. second) marginal of $\pi$.

The Wasserstein distance is a natural metric to compare discrete and absolutely continuous probability distributions and its use for distributional robustness has attracted a lot of attention; see e.g., (Shafieezadeh Abadeh et al., 2015; Sinha et al., 2018; Shafieezadeh-Abadeh et al., 2019; Li et al., 2020; Kwon et al., 2020) and the review articles (Blanchet et al., 2021; Kuhn et al., 2024).

---

[3]While this motivation is valid and provides a good intuition, we note that having $\mathrm{P_\star} \in \mathcal{U}_\rho(\mathrm{P_n})$ may be slightly too demanding compared to verifying directly (11). We will get back to this point in more details in the next sections.

[4]We focus here on the type-2 distance for simplicity. Actually, the transport cost could be easily modified in several parts of the paper to accommodate for instance discrete sets for classification problems. The results still hold without modifications, except for Lemma 4 which has to be modified accordingly (see Gao and Kleywegt (2023, Rem. 2)).

### 3.2 Construction of WDRO risk-controlled prediction sets

Now, let us construct risk-controlled prediction sets using Wasserstein distributionally robust optimization problems using the rationale of Section 2.2.

#### 3.2.1 Preliminaries

In this section, we will use as an upper-bound to the true risk the Wasserstein distributionally robust empirical risk defined as

$$\widehat{\mathcal{R}}_n^\rho(\lambda) := \sup_{Q \in \mathcal{P}(\Xi): W_2(P_n, Q) \le \rho} \mathbb{E}_{(X,Y) \sim Q}[L(Y, \mathcal{T}_\lambda(X))]. \tag{12}$$

Now, in order for this proxy to have desired properties such as finiteness, we have to introduce adequate assumptions.

**Assumption 3.** *Assume that $\widehat{\mathcal{R}}_n(\lambda) \le +\infty$ for all $\lambda \in \Lambda$ and that either of the following properties hold:*

*a)* *the set $\mathcal{X} \times \mathcal{Y}$ is bounded and the function $(x, y) \mapsto L(y, \mathcal{T}_\lambda(x))$ is upper-bounded uniformly in $\lambda$*

*b)* *the set $\mathcal{X} \times \mathcal{Y}$ is unbounded and there exists finite constants $M, M' > 0$, independent of $\lambda$, such that $L(y, \mathcal{T}_\lambda(x)) \le M + M'(\|x\|^{2-\varepsilon} + \|y\|^{2-\varepsilon})$ for some $\varepsilon \in (0, 2)$*

These assumptions are rather mild as the conformal losses are usually bounded (especially in light of Assumption 2-iii)). This implies that $\widehat{\mathcal{R}}_n^\rho(\lambda)$ is finite by classical results of the WDRO literature (e.g., Gao and Kleywegt (2023)).

**Lemma 4.** *Let Assumption 3 hold. Then, for all $\lambda \in \Lambda$, and all $\rho \ge 0$, $\widehat{\mathcal{R}}_n^\rho(\lambda) < +\infty$.*

*Proof.* This directly follows from Gao and Kleywegt (2023, Th. 1) as Assumption 3 guarantees that $(x, y) \mapsto L(y, \mathcal{T}_\lambda(x)) \in L^1(P_n)$ and that the growth rate (see Gao and Kleywegt (2023, Def. 4)) is finite. $\square$

#### 3.2.2 Step A

In order to build upper-bounds of the form of (7), one has to guarantee that $\widehat{\mathcal{R}}_n^\rho(\lambda)$ is a valid upper-bound in our goal provided that the robustness radius is chosen adequately. Since we consider a Wasserstein distributionally robust upper bound, we can directly rely on measure concentration results in $W_2$ distance (e.g., Fournier and Guillin (2015)). This naturally provides a candidate radius that has the merit to be valid uniformly in $\lambda$ (and in the base model) as it gives a probability for $P_0$ to be included in the constraint set in the definition of $\widehat{\mathcal{R}}_n^\rho(\lambda)$ in (12).

**Lemma 5.** *Suppose that $\mathbb{E}_{\xi \sim P_0}\big[\exp(\gamma \|\xi\|^\beta)\big] < \infty$ for some $\beta > 2, \gamma > 0$. For any $\delta \in (0, 1)$, there exists $\rho(n, \delta) = \mathcal{O}\big((n^{-1} \log(\delta^{-1}))^{\frac{4}{\max(4,d)}}\big)$ so that for any $\rho \ge \rho(n, \delta)$ we have*

$$\mathbb{P}_{(X_i, Y_i)_{i=1}^n \sim P_0^{\otimes n}}\Big[\forall \lambda \in \Lambda, \quad \mathcal{R}(\lambda) \le \widehat{\mathcal{R}}_n^\rho(\lambda)\Big] \ge 1 - \delta \tag{13}$$

*whenever $W_2(P_0, P_\star) \le \rho - \rho(n, \delta)$. In particular, if $P_0 = P_\star$, Eq. (13) holds as soon as $\rho \ge \rho(n, \delta)$.*

*Proof.* From Fournier and Guillin (2015, Th. 2), case (1), we have that if $\mathcal{E}_{\beta,\gamma} := \mathbb{E}_{\xi \sim P_0}\big[\exp(\gamma \|\xi\|^\beta)\big] < \infty$ for some $\beta > 2, \gamma > 0$, then for any $x > 0$

$$\mathbb{P}_{(X_i, Y_i)_{i=1}^n \sim P_0^{\otimes n}}[W_2(P_n, P_0) < x] \ge 1 - C \begin{cases} \exp(-c\,n\,x) & \text{if } d < 4 \quad \text{and } x \le 1 \\ \exp(-c\,n\,x/\log(2 + 1/\sqrt{x})^2) & \text{if } d = 4 \quad " \\ \exp(-c\,n\,x^{d/4}) & \text{if } d > 4 \quad " \\ \exp(-c\,n\,x^{\beta/4}) & \text{otherwise} \end{cases} \tag{14}$$

where constants $c$ and $C$ only depend on $d$, $\beta$, $\gamma$ and $\mathcal{E}_{\beta,\gamma}$.

Fix $\delta \in (0,1)$. Then, with

$$
\rho(n,\delta) := \begin{cases}
\frac{\log(C/\delta)}{cn} & \text{if } d \leq 4 \qquad \text{and } n \geq \frac{\log(C/\delta)}{c} \\
\left(\frac{\log(C/\delta)}{cn}\right)^{4/d} & \text{if } d > 4 \qquad\qquad\quad \text{''} \\
\left(\frac{\log(C/\delta)}{cn}\right)^{4/\beta} & \text{otherwise}
\end{cases} \quad ,
$$

we have $\mathbb{P}_{(X_i,Y_i)_{i=1}^n \sim P_0^{\otimes n}}[W_2(P_n, P_0) < \rho(n,\delta)] \geq 1 - \delta$.

In turn, if $W_2(P_0, P_\star) \leq \rho - \rho(n,\delta)$, then with probability $1 - \delta$, $W_2(P_\star, P_n) \leq \rho$ and thus

$$
\forall \lambda \in \Lambda, \quad \mathcal{R}(\lambda) = \mathbb{E}_{(X,Y) \sim P_\star}[L(Y, \mathcal{T}_\lambda(X))] \leq \sup_{Q:W_2(Q,P_n) \leq \rho} \mathbb{E}_{(X,Y) \sim Q}[L(Y, \mathcal{T}_\lambda(X))] = \widehat{\mathcal{R}}_n^\rho(\lambda)
$$

which is the claimed result. $\qquad\square$

This result validates the use of WDRO for Step A of the production of risk-controlled prediction sets. One disadvantage of the approach is that the choice of the uncertainty radius is not easy to choose as it depends non-trivially on the problem's parameters.

**Remark 6** (Conformal guarantees on the WDRO problem)**.** *By flipping the problem, the above result means that for any choice of radius $\rho > 0$, the* Wasserstein distributionally robust conformal prediction problem (12) *provides a natural guarantee of the form of Eq.* (13) *with probability $1 - \delta$ where $\delta = C \exp(-c\,n\,\rho^p)$ with $p \in [1, \max(d,\beta)/4]$ depending on the value of $d$ and $\rho$ (see Eq.* (14)*).*

### 3.2.3 Step B

Now, we want to find some $\hat{\lambda}_n$ so that (8) is satisfied with our WDRO upper-bound. In words, we have to make the conformal prediction set sufficiently large so the *robust* conformal loss is arbitrarily small. We first show existence of such a $\hat{\lambda}_n$.

**Lemma 7** (Existence)**.** *Let Assumption 2 hold. Then, for any $\alpha \in (0,1)$ and $\rho > 0$ there exists $\overline{\lambda}$ satisfying*

$$
\forall \lambda \in \Lambda, \quad \lambda \geq \overline{\lambda} \Rightarrow \widehat{\mathcal{R}}_n^\rho(\lambda) \leq \alpha.
$$

*Proof.* Fix $\alpha \in (0,1)$. Using Assumption 2-iii), there exists some $\overline{\lambda}$ such that for all $\lambda \geq \overline{\lambda}$, $L(y, \mathcal{T}_\lambda(x)) \leq \alpha$ for all $(x,y) \in \mathcal{X} \times \mathcal{Y}$.

Now, we have that for any $\rho > 0$, and for all $\lambda \geq \overline{\lambda}$,

$$
\begin{aligned}
\widehat{\mathcal{R}}_n^\rho(\lambda) &= \sup_{Q \in \mathcal{P}(\Xi):W_2(P_n,Q) \leq \rho} \mathbb{E}_{(X,Y) \sim Q}[L(Y, \mathcal{T}_\lambda(X))] \\
&\leq \alpha
\end{aligned}
$$

where we simply used that the supremum is taken over probability distributions. $\qquad\square$

The result is thus almost direct from Assumption 2-iii) nevertheless it tells us that under this assumption (which holds in most cases of interest), one can find a suitable $\hat{\lambda}_n$ for any $\alpha$, which is not the case with the classical Hoeffding or Bernstein bounds (see the upcoming Fig. 2 in Section 4.1).

The above result means that one can find a data-independent value for $\hat{\lambda}_n$ to get (8). Nevertheless, in practice, we shall rely on data-driven values both for tractability and performance. In order to so, a desirable property is monotonicity in $\lambda$, as it enables the use of dichotomy search.

**Lemma 8** (Monotonicity)**.** *Let Assumption 2 hold. Then, for any $\rho > 0$, $\widehat{\mathcal{R}}_n^\rho(\lambda)$ is monotonically decreasing.*

*Proof.* Fix $\rho > 0$. Using Assumption 2-i) and ii), we have that if $\lambda_1 \geq \lambda_2$, $L(y, \mathcal{T}_{\lambda_1}(x)) \geq L(y, \mathcal{T}_{\lambda_2}(x))$ for all $(x, y) \in \mathcal{X} \times \mathcal{Y}$. Thus,

$$
\begin{aligned}
\widehat{\mathcal{R}}_n^\rho(\lambda_2) &= \sup_{Q \in \mathcal{P}(\Xi): W_2(P_n, Q) \leq \rho} \mathbb{E}_{(X,Y) \sim Q}[L(Y, \mathcal{T}_{\lambda_2}(X))] \\
&\leq \sup_{Q \in \mathcal{P}(\Xi): W_2(P_n, Q) \leq \rho} \mathbb{E}_{(X,Y) \sim Q}[L(Y, \mathcal{T}_{\lambda_1}(X))] = \widehat{\mathcal{R}}_n^\rho(\lambda_1)
\end{aligned}
$$

hence the monotonocity. $\qquad\square$

### 3.2.4 Conclusion

We are now in position to state our main result on the construction of risk-controlled prediction sets using WDRO. It states that if the radius $\rho$ is well chosen, there exists $\hat{\lambda}_n$ that gives the sought guarantees and more precisely that any that satisfies $\widehat{\mathcal{R}}_n^\rho(\hat{\lambda}_n) \leq \alpha$ will do thanks to the monotonicity.

**Theorem 9.** *Let Assumptions 2 and 3 hold and suppose that $\mathbb{E}_{\xi \sim P_0}\left[\exp(\gamma \|\xi\|^\beta)\right] < \infty$ for some $\beta > 2, \gamma > 0$. For any $\alpha \in (0, 1)$ and any $\delta \in (0, 1)$, there exists $\rho(n, \delta) = \mathcal{O}\left((n^{-1}\log(\delta^{-1}))^{\frac{4}{\max(4,d)}}\right)$ so that for any $\rho \geq \rho(n, \delta)$, choosing $\hat{\lambda}_n$ so that $\widehat{\mathcal{R}}_n^\rho(\hat{\lambda}_n) \leq \alpha$, we have*

$$
\mathbb{P}_{(X_i, Y_i)_{i=1}^n \sim P_0^{\otimes n}}\left[\boldsymbol{\mathcal{R}}(\hat{\lambda}_n) \leq \alpha\right] \geq 1 - \delta
$$

*whenever $W_2(P_0, P_\star) \leq \rho - \rho(n, \delta)$. In particular, if $P_0 = P_\star$, the result holds as soon as $\rho \geq \rho(n, \delta)$.*

The result directly follows from Lemmas 5, 7 and 8. Although quite natural once the adequate results of the WDRO literature and assumptions are put together, it highlights that the the connection between WDRO and RCPS (or possibly other conformal methods) can be of interest. After discussing further the reach and limitations of this result, the remaining of the paper will thus be devoted to the computation and numerical evaluation of this methodology.

## 3.3 Discussion

The main parameter to tune in WDRO is the robustness radius $\rho$ (one could also mention the transport cost), the same holds for the approach developed here. One point to notice is that once $\rho$ is chosen, the statistical guarantee of Eq. (13) is uniform in $\lambda$. The sole condition on it in Theorem 9 is that it should be greater than some prescribed radius $\rho(n, \delta)$. Indeed, the excess $\rho - \rho(n, \delta)$ can be interpreted as the amount of distribution shift that can be handled by this approach. This point is important as it means that WDRO risk-controlled prediction sets are *inherently robust to some amount of distribution shift*.

Nevertheless, a drawback is that $\rho(n, \delta)$ is not explicitly computable. In fact, the tuning of $\rho$ is still an active research direction in the WDRO community. A common choice is $\propto 1/\sqrt{n}$, reflecting the fact that the radius should be larger in a data-poor environment. However, the amount of distribution shift between the training distribution and the true one is also difficult to estimate in general so the drawback is not really specific to the developed approach.

Another limitation is that the prescribed radius $\rho(n, \delta)$ in Theorem 9 (coming from Lemma 5) suffers from the curse of the dimensionality. Indeed, $\rho(n, \delta)$ decreases in $\mathcal{O}(1/n^{1/d})$ which can be very slow when $d$ is large. This is due to the fact that we concentrate here the full probability distribution instead of looking directly at the WDRO problem. Indeed, for WDRO problems, a scaling in $\mathcal{O}(1/\sqrt{n})$ appears to be asymptotically optimal (Blanchet et al., 2022; Blanchet and Shapiro, 2023). Recent works have managed to demonstrate this formally for a wide class of WDRO objectives (see (Azizian et al., 2023b; Le and Malick, 2024)); nevertheless, the required assumptions on the loss make the results inapplicable in our setting of interest. In order to focus on the main points of the paper, we leave this question open.

## 3.4 Dual problem and Numerical Tractability

As problem (12) is an infinite dimensional problem over a space of measures, duality has been a central tool in both the theoretical analyses and computational schemes of WDRO from the onset (Shafieezadeh Abadeh

et al., 2015; Esfahani and Kuhn, 2018). Indeed, the dual problem (Gao and Kleywegt, 2023; Sinha et al., 2018; Blanchet and Murthy, 2019) can be written as a one-dimensional problem involving $n$ subproblems:

$$\widehat{\mathcal{D}}_n^\rho(\lambda) := \inf_{\gamma \geq 0} \gamma \rho^2 + \mathbb{E}_{(X,Y) \sim \mathrm{P_n}} \left[ \sup_{(x',y') \in \Xi} \left\{ L(y', \mathcal{T}_\lambda(x')) - \gamma \|X - x'\|^2 - \gamma \|Y - y'\|^2 \right\} \right]. \tag{15}$$

Though its expression appears simpler, the presence of a sup makes (15) involved to solve in general. Nevertheless, since both the optimization in $\gamma$ and in the supremum are finite-dimensional, it open the door to easier practical implementation and carries important information about the solution of the primal problem.

### 3.4.1 Strong duality and worst-case distributions

The following theorem guarantees that we have strong duality i.e., that the optimal value of the dual problem corresponds to $\widehat{\mathcal{R}}_n^\rho(\lambda)$ which is the workhorse of our approach.

**Theorem 10** (Strong duality). *Under Assumptions 2 and 3, for all $\lambda \in \Lambda$,*

$$\widehat{\mathcal{R}}_n^\rho(\lambda) = \widehat{\mathcal{D}}_n^\rho(\lambda)$$

*and are finite. Furthermore, the worst case distribution for the WDRO risk is attained and*

$$\mathrm{Q}^\star := \operatorname*{arg\,sup}_{\mathrm{Q} \in \mathcal{P}(\Xi) : W_2(\mathrm{P_n}, \mathrm{Q}) \leq \rho} \mathbb{E}_{(X,Y) \sim \mathrm{Q}}[L(Y, \mathcal{T}_\lambda(X))]$$

*is supported on $n+1$ atoms and $\mathrm{Q}^\star = \frac{1}{n} \sum_{i=1}^n \delta_{\xi_i^\star} + p_0(\delta_{\xi_{i_0}^\square} - \delta_{\xi_{i_0}^\star})$ where*

$$\xi_i^\star \in \operatorname*{arg\,sup}_{(x',y') \in \Xi} \left\{ L(y', \mathcal{T}_\lambda(x')) - \frac{\gamma^\star}{2} \|X_i - x'\|^2 - \frac{\gamma^\star}{2} \|Y_i - y'\|^2 \right\}, \quad i = 1, .., n$$

*with $\gamma^\star$ the optimal dual variable in (15), $i_0 \in \{1, .., n\}$, $p_0 \in [0, 1]$,*
*and $\xi_{i_0}^\square \in \operatorname*{arg\,sup}_{(x',y') \in \Xi} \left\{ L(y', \mathcal{T}_\lambda(x')) - \frac{\gamma^\star}{2} \|X_{i_0} - x'\|^2 - \frac{\gamma^\star}{2} \|Y_{i_0} - y'\|^2 \right\}.$*

*Proof.* As for Lemma 4, the strong duality follows from Gao and Kleywegt (2023, Th. 1). Furthermore, we can invoke Yue et al. (2022, Th. 4) to get the existence of a worst case distribution in $\widehat{\mathcal{R}}_n^\rho(\lambda)$ and Gao and Kleywegt (2023, Cor. 2) to obtain the form of the worst case distribution. □

This result gives an intuition on what the worst distribution shift looks like. In words, the worst case distribution is a shift of the empirical one towards the regions where the conformal loss is large.

### 3.4.2 A special case

To give a simple example where (15) leads to an explicit solution, consider the setup of Example 2.1 where $\mathcal{T}_\lambda(x) = \{y \in \mathcal{Y} : |y - f_\theta(x)| < \lambda\}$ and $L(y, \mathcal{T}_\lambda(x)) = \mathbb{1}_{y \notin \mathcal{T}_\lambda(x)} = \mathbb{1}_{|y - f_\theta(x)| \geq \lambda}$. In addition, let us suppose that only the $Y$ are moved in the worst case distribution (this can be seen for instance as the transport cost for $X$ being multiplied by a large constant).

The dual problem is then

$$\widehat{\mathcal{D}}_n^\rho(\lambda) := \inf_{\gamma \geq 0} \gamma \rho^2 + \frac{1}{n} \sum_{i=1}^n \left[ \sup_{y' \in \mathbb{R}} \left\{ \mathbb{1}_{|y' - f_\theta(X_i)| \geq \lambda} - \gamma \|Y_i - y'\|^2 \right\} \right] \tag{16}$$

and notice that for each $i$:

1. if $|Y_i - f_\theta(X_i)| \geq \lambda$ (the data point is *not* in the confidence region), then $\sup_{y' \in \mathbb{R}} \left\{ \mathbb{1}_{|y' - f_\theta(X_i)| \geq \lambda} - \gamma \|Y_i - y'\|^2 \right\} = 1$, attained by $Y_i$;

$$f_\theta(X_i) - \lambda \qquad\qquad f_\theta(X_i) \qquad\qquad f_\theta(X_i) + \lambda$$

$$\cdots\cdots\cdots\cdots\vdash\text{-}\text{-}\text{-}\text{-}\text{-}\text{-}\vdash\!\!\!\!\!-\!\!\!-\!\!\!-\!\!\!+\!\!\!-\!\!\!-\!\!\!-\!\!\!\vdash\text{-}\text{-}\text{-}\text{-}\text{-}\text{-}\dashv\cdots\cdots\cdots\cdots$$

$$f_\theta(X_i) - \lambda + \tfrac{1}{\sqrt{\gamma}} \qquad\qquad\qquad f_\theta(X_i) + \lambda - \tfrac{1}{\sqrt{\gamma}}$$

**Figure 1:** Illustration of the value of the suprema in (16) depending on $Y_i$. If $Y_i$ is in the plain region around $f_\theta(X_i)$ (Case 2.a), it is sufficiently well classified (depending on $\lambda$ but also $\gamma$ and thus $\rho$ as show after) so it is too costly to displace it, the optimal $y'$ is $Y_i$ and the supremum is worth 0. If $Y_i$ is in the dashed region (Case 2.b), it is well classified but can be pushed outside the confidence region, the optimal $y'$ is then $f_\theta(X_i) \pm \lambda$ and the supremum is in $(0, 1)$. Finally, if $Y_i$ is in the dotted region (Case 1), no displacement is necessary to be out of the confidence region and thus the optimal $y'$ is $Y_i$ and the supremum is worth 1.

2. otherwise (the data point is in the confidence region), in order to push $y'$ outside of the prediction interval, it has to be at distance at least $\lambda - |Y_i - f_\theta(X_i)| > 0$ of $Y_i$ and thus:

   a) if $|Y_i - f_\theta(X_i)| > \lambda - \frac{1}{\sqrt{\gamma}} \Leftrightarrow 1 - \gamma(\lambda - |Y_i - f_\theta(X_i)|)^2 > 0$, the displaced point leads to a positive value for the term in braces. Thus, it is optimal to displace the point and the supremum is equal to $1 - \gamma(\lambda - |Y_i - f_\theta(X_i)|)^2 > 0$;

   b) otherwise, $|Y_i - f_\theta(X_i)| \le \lambda - \frac{1}{\sqrt{\gamma}} \Leftrightarrow 1 - \gamma(\lambda - |Y_i - f_\theta(X_i)|)^2 \le 0$, the displaced point leads to a negative value for the term in braces. Thus, it is optimal to take $y' = Y_i$, the supremum is then equal to 0;

This reasoning is illustrated by Fig. 1; note that we are actually exhibiting the worst-case distribution. Without loss of generality, we reorder the samples by decreasing error so that

$$|Y_1 - f_\theta(X_1)| \ge |Y_2 - f_\theta(X_2)| \ge \cdots \ge |Y_n - f_\theta(X_n)|$$

and let $i_0 = \max\{i : |Y_i - f_\theta(X_i)| \ge \lambda\}$ and $i_\star^\gamma = \max\{i : |Y_i - f_\theta(X_i)| \ge \lambda - \frac{1}{\sqrt{\gamma}}\}$ (so that indices in $[1, i_0]$ correspond to Case 1, indices in $[i_0 + 1, i_\star^\gamma]$ correspond to Case 2.a, indices in $[i_\star^\gamma + 1, n]$ correspond to Case 2.b) to have

$$\widehat{\mathcal{D}}_n^\rho(\lambda) = \inf_{\gamma \ge 0} \gamma \rho^2 + \frac{1}{n}\left(\sum_{i=1}^{i_0} 1 + \sum_{i=i_0+1}^{i_\star^\gamma}\left[1 - \gamma(\lambda - |Y_i - f_\theta(X_i)|)^2\right] + \sum_{i=i_\star^\gamma+1}^{n} 0\right)$$

$$= \inf_{\gamma \ge 0} \gamma \rho^2 + \frac{i_0}{n} + \frac{1}{n}\sum_{i=i_0+1}^{i_\star^\gamma}\left[1 - \gamma(\lambda - |Y_i - f_\theta(X_i)|)^2\right] \tag{17}$$

We are left with finding the optimal value of the dual variable $\gamma$. In fact, by looking at the definition of $i_\star^\gamma$ and the expression of (17), it is direct that $\gamma$ can be sought as $1/(\lambda - |Y_i - f_\theta(X_i)|)^2$ for $i$ in $[i_0 + 1, n]$. By computing the value of (17) for these $n - i_0 - 1$ points and denoting by $i_\star$ the arg min, we obtain an optimal dual variable $\gamma^\star = 1/(\lambda - |Y_{i_\star} - f_\theta(X_{i_\star})|)^2$ and we have

$$\widehat{\mathcal{R}}_n^\rho(\lambda) = \widehat{\mathcal{D}}_n^\rho(\lambda) = \frac{\rho^2}{(\lambda - |Y_{i_\star} - f_\theta(X_{i_\star})|)^2} + \frac{i_\star}{n} - \frac{1}{n}\sum_{i=i_0+1}^{i_\star}\frac{(\lambda - |Y_i - f_\theta(X_i)|)^2}{(\lambda - |Y_{i_\star} - f_\theta(X_{i_\star})|)^2} \tag{18}$$

and the worst case distribution is exactly the training data except for the indices between $i_0 + 1$ and $i_\star$ (included), corresponding some points that used to fall into the conformal prediction set and that are push at the boundary. The formula in (18) seems difficult to interpret in general but we nevertheless see some interesting similarities in its formulation with the Waudby-Smith–Ramdas bound (Waudby-Smith and Ramdas, 2024; Bates et al., 2021) despite the completely different approaches in their construction.

**Remark 11.** *In order to include variations of $X$ while keeping a closed form, it would be possible to consider a linear model $f_\theta(X) = \langle \theta, X \rangle$ and apply the same reasoning as above. Nevertheless, the derivations become rapidly cumbersome and the result is not much more informative.*

### 3.4.3 Regularization

Providing explicit expressions for the the solution of (15) as in the previous section is out of reach for many situations of interest, a common limitation of WDRO. Nevertheless, entropic regularization have of WDRO have been proposed and studied recently, offering promising performances (Azizian et al., 2023a;b; Wang et al., 2023). These approaches can be directly used in our context; by adding an entropy regularization term to the transportation problem, the dual problem (15) becomes

$$\widehat{\mathcal{D}}_n^{\rho,\varepsilon}(\lambda) := \inf_{\gamma \geq 0} \gamma \rho^2 + \varepsilon \, \mathbb{E}_{(X,Y) \sim \mathrm{P_n}} \left[ \log \mathbb{E}_{(X',Y') \sim \mathrm{Q}((X,Y))} \exp \left\{ \frac{L(Y', \mathcal{T}_\lambda(X')) - \gamma \|X - X'\|^2 - \gamma \|Y - Y'\|^2}{\varepsilon} \right\} \right]$$
(19)

where $\varepsilon > 0$ is the regularization strength, and $\mathrm{Q}((X,Y))$ is a user-defined conditional distribution. Typical choices are $\varepsilon = \rho$ and $\mathrm{Q}((X,Y)) = \mathcal{N}((X,Y), \rho I)$.

The regularized dual (19) enables to find an approximate solution of (15) by relying on gradient-based methods (since the supremum in (15) is replaced by a differentiable function). Nevertheless, one still has to sample the points $(X', Y')$ in order to numerically approximate the inner expectation. Compared to the special case of the previous section or to classical conformal bounds, the numerical cost of solving this problem is clearly much higher (due to the sampling and optimization in $\gamma$) but when jointly training and performing conformal prediction on a model, the computation times are on par with classical adversarial training methods. We refer to the papers above for more details and to Vincent et al. (2024) for a practical implementation.

### 3.5 Simultaneous learning and conformal prediction

A crucial point in our analysis is that the statistical guarantees for WDRO are uniform in the loss functions; in Lemma 5, the result is uniform in $\lambda$. This means that if the prediction depends on some parameter $\theta$, we can learn $\theta$ while constructing our WDRO conformal prediction sets, using the same dataset. This feature is seldom found in the conformal prediction literature where the model and conformal prediction set usually have to be learned using separate datasets (a recent exception is (Braun et al., 2025)).

To make this more precise, consider the setup of Example 2.1 where $\mathcal{T}_\lambda(x) = \{y \in \mathcal{Y} : |y - f_\theta(x)| < \lambda\}$ and $L(y, \mathcal{T}_\lambda(x)) = \mathbb{1}_{y \notin \mathcal{T}_\lambda(x)} = \mathbb{1}_{|y - f_\theta(x)| \geq \lambda}$. Then, we can use

$$\widehat{\mathcal{R}}_n^\rho(\lambda, \theta) = \sup_{\mathrm{Q} \in \mathcal{P}(\Xi) : W_2(\mathrm{P_n}, \mathrm{Q}) \leq \rho} \mathbb{E}_{(X,Y) \sim \mathrm{Q}} \left[ \mathbb{1}_{|y - f_\theta(x)| \geq \lambda} \right].$$

Then, choosing $\hat{\lambda}_n, \hat{\theta}_n$ so that $\widehat{\mathcal{R}}_n^\rho(\hat{\lambda}_n, \hat{\theta}_n) \leq \alpha$, we have the same guarantees as Theorem 9. Thus, we can actually optimize the model jointly with the risk-controlled prediction set estimation, *on the same dataset*. Note here that the obtained model would be different from the one obtained by empirical risk minimization of even WDRO as it is learned to optimize the *conformal* loss. Numerically, a good solution is to rely on the regularized formulation mentioned in Section 3.4.3.

## 4 Numerical illustrations

In this section, we illustrate the results of the paper and the applicability of WDRO-based Risk Controlling Prediction Sets. Our goal is thus to show how the WDRO-based bounds behave compared to classical approaches for RCPS and how they can encompass distribution shifts and simultaneous training and conformal prediction, rather than performing a complete performance evaluation. The code used for these experiments is available at https://github.com/iutzeler/rcps-wdro.

We place ourselves in the regression setting of Example 2.1 and compare the following upper-bound functions $\widehat{\mathcal{R}}_n^+(\lambda)$ aimed at verifying (7) in Step A:

- *Hoeffding:* $\widehat{\mathcal{R}}_n^H(\lambda) = \widehat{\mathcal{R}}_n(\lambda) + \sqrt{\frac{1}{2n} \log\left(\frac{1}{\delta}\right)}$ (see (Bates et al., 2021, Sec. 3.1.1))

- *Bernstein:* $\widehat{\mathcal{R}}_n^B(\lambda) = \widehat{\mathcal{R}}_n(\lambda) + \hat{\sigma}\sqrt{\frac{2}{n}\log\left(\frac{2}{\delta}\right)} + \frac{7}{3(n-1)}\log\left(\frac{2}{\delta}\right)$ where $\hat{\sigma}^2$ is the standard unbiased variance estimator for the losses (see (Bates et al., 2021, Sec. 3.1.3))

- *Simple WDRO:* The bound of (18)

- *SKWDRO:* The bound of (19) where the 0/1 loss is replaced by a smooth version[5] solved using the package skwdro (Vincent et al., 2024)

In all the section, unless otherwise specified, we take $\alpha = 0.1$ and $\delta = 0.05$. For the WDRO-based approaches, we take $\rho = c/\sqrt{n}$ with $c = \sqrt{2}$ for *SKWDRO* and $c = 10\sqrt{2}$ for *Simple WDRO*. Finally, note that for both WDRO-based approaches, the solved objectives ((18) and (19)) are dual-based.

**Remark 12** (Choice of the radius $\rho$). *The choice of the radius is an important and difficult question in distributionally robust optimization both theoretically and practically. We can distinguish two approaches: (i) generic measure concentration (as used in Esfahani and Kuhn (2018) and Lemma 5), which gives $\rho = (log(C/\delta)/(cn))^{4/d}$ and thus suffers from the curse of dimensionality (but can be used directly to obtain statistical guarantees on WDRO estimators); or (ii) specific concentration on the WDRO (dual) objective which give $\rho = C'/\sqrt{n}$ but are unfortunately elusive for the conformal objective functions considered in the paper (mainly due to the discontinuity, see Azizian et al. (2023b)). In both cases, constants are not explicit.*

*Nevertheless, as in classical WDRO we found that $\rho \propto 1/\sqrt{n}$ was performing rather well all around and this was used in our experiments as mentioned above. This finding is in line with the "practical rate" for WDRO estimation as mentioned in Sec. 7.2.D of Esfahani and Kuhn (2018).*

*Furthermore, for a specific dataset, several approaches can be used to tune the radius adequately. Indeed, as training and conformal prediction can be done simultaneously over the same dataset (contrary to most conformal prediction techniques which rely on a separate calibration set), a small portion of the data can be held out for cross validating the value of $\rho$ to satisfy the requirements with virtually no performance loss over the whole training procedure. This approach is detailed in (Esfahani and Kuhn, 2018, Sec. 7.2.B,C) for instance and can be an alternative to the standard $1/\sqrt{n}$ choice.*

*Finally, in the case of an unknown distribution shift, these techniques could be too optimistic (unless some samples from the shifted distribution are available) and the (square of) the radius should be theoretically increased by the measure of the shift (in $W_2^2$ distance). In practice, this is often too conservative as the geometry of the Wasserstein distance with Euclidean cost may not capture adequately the loss degradation. This is a general issue in the modeling of distribution shifts that goes beyond the scope of the present paper.*

## 4.1 Coverage and interval size

In this section, our experimental setup is the following. We use 1000 samples generated from the make-regression function of scikit-learn:[6] 500 are used to train a linear prediction model (using scikit-learn's LinearRegression estimator), 250 are used for calibration (which is $n$ in the notation of the paper), 250 are used for evaluation.

In Fig. 2, we display the values of the considered upper-bounds as a function of the prediction size $\lambda$ for one realization of the experiment. The selected prediction size $\hat{\lambda}_n$ for each upper-bound is the smallest value of $\lambda$ for which the upper-bound is below the target risk (represented as a dashed line), the smaller the better as this will result in tighter prediction sets for the same guarantee. We observe that WDRO-based upper-bounds are consistently smaller than Hoeffding or Bernstein meaning they will lead to tighter prediction sets in general.

In Fig. 3, we repeat our experiment 100 times and report boxplots of the coverages (i.e., one minus the risk) evaluated on the test data as well as prediction interval sizes. We observe that WDRO-based approaches lead to admissible coverages (slightly closer to the target risk) with smaller interval sizes. Thus, the constitute interesting substitutes for Hoeffding or Bernstein-based approaches.

---

[5]Precisely, we replace $\mathbb{1}_{u>0}$ by $\frac{1}{1+\exp(-30u-3)}$.

[6]We use the options n_features=5, n_informative=3, noise=20 leading to a problem in dimension $d = 5$ with a fair amount of noise.

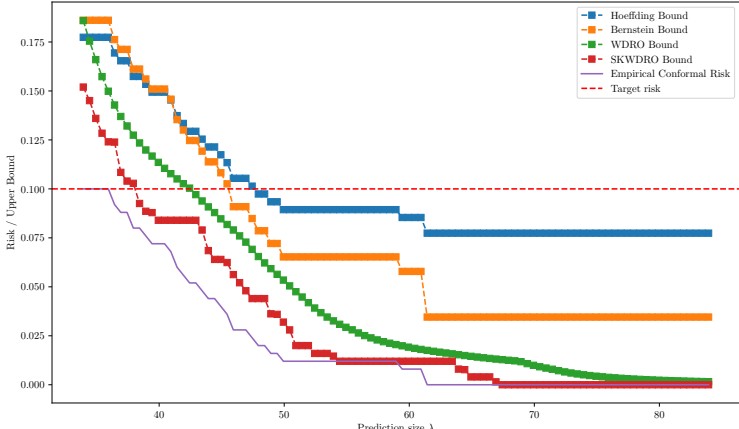

**Figure 2:** Behavior of the RCPS bounds a function of $\lambda$

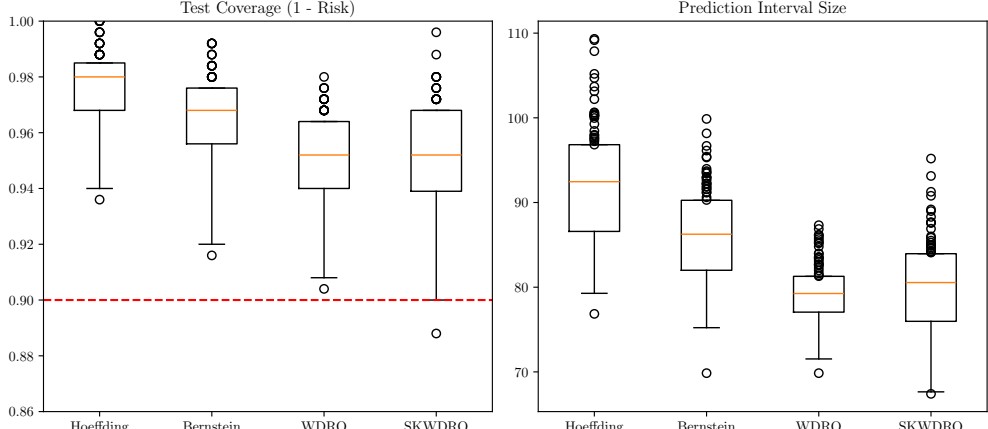

**Figure 3:** Performance of the RCPS bounds over $N = 100$ runs. The dashed line represents the target coverage $1 - \alpha$. The whiskers of the boxplot are set to the $\delta$ and $1 - \delta$ quantiles so that the target of RPCS is to have a lower whisker above the dashed line.

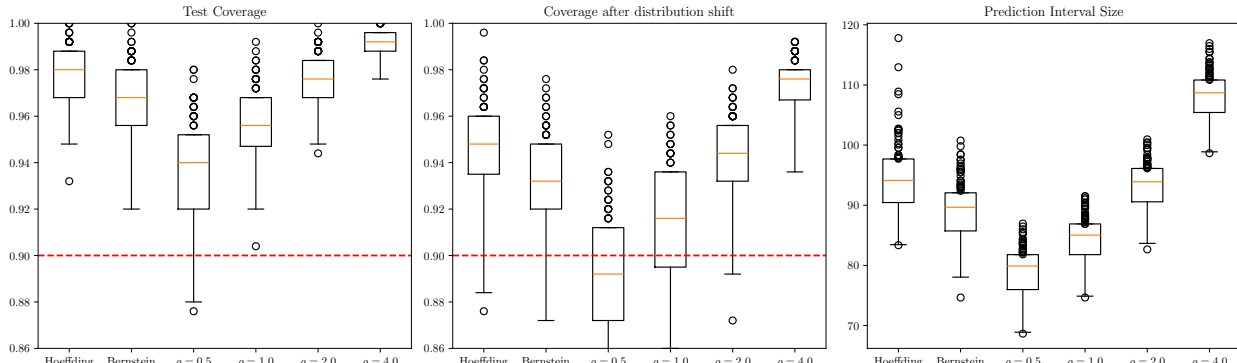

**Figure 4:** Performance of the RCPS bounds over $N = 100$ runs *with and without distribution shift*. The dashed line represents the target coverage $1 - \alpha$. The whiskers of the boxplot are set to the $\delta$ and $1 - \delta$ quantiles so that the target of RPCS is to have a lower whisker above the dashed line. The leftmost figure corresponds to the classical test coverage (as in the previous section) while the center one corresponds to the coverage on the shifted dataset. For the *WDRO* bound, the performance for several values of $\rho$ are displayed

## 4.2 Distribution shifts

We place ourselves in the same setup as before. To simulate a distribution shift, we create a shifted dataset by copying the test dataset and adding to the points $(X_i)$ a Gaussian noise of mean $(0.25, 0, 0, 0, 0)$ and covariance $0.25^2 I$, resulting in an Wasserstein distance of $0.612$ between testing and training data. In Fig. 4, we display the coverage for the test set and the shifted set. We compare different values of $\rho$ for the WDRO bound. We notice first that the Bernstein and Hoeffding bounds are so pessimistic that they can actually accommodate for quite large distribution shifts in practice (without theoretical grounding though). For the WDRO-based bounds, we see that, as expected, the interval sizes and coverage grow with $\rho$, which enables to take into account different magnitudes of shift (we recall that following Theorem 9, a WDRO model can accommodate for a shift of magnitude $s$ in Wasserstein distance whenever $\rho \geq s + \rho(n, \delta)$ where $\rho(n, \delta)$ is the concentration radius that can be taken $\propto 1/\sqrt{n}$).[7] In particular, in the present case, the shift is of magnitude $s = 0.612$ and the "practical concentration radius" is $\rho_n = 10\sqrt{2/250} = 0.894$ leading to a minimal radius of $1.506$. Hence, the choice $\rho = 0.5$ is smaller than the distribution shift magnitude and thus does not provide coverage in the shifted situation (middle plot) and even without shift (left plot). The choice $\rho = 1.0$ is closer to the prescribed value, provides a correct coverage before shift but does not match the RCPS target after shift. A satisfying performance after shift is attained with $\rho = 2.0$. This validates the interest of this approach for directly integrating distribution shifts.

## 4.3 Simultaneous training and conformal inference

In this section, we draw $n = 200$ samples from the following distribution: $X$ is drawn uniformly in $[-2, 2]$ and $Y = f(X) + U$ where $f(t) = \frac{10}{\exp(t) + \exp(-t)} + t$ and $U$ is a standard Gaussian variable.

For a fixed $\lambda$, we perform a degree-4 polynomial regression of $Y$ from $X$ and compare two approaches for training and conformal inference using the *SKWDRO* bound:[8]

- *Separate.* The regression is learned using `sklearn`'s Linear Regression on polynomial features then the conformal upper bound is computed.

- *Joint.* The polynomial model is trained to minimize the conformal loss, i.e., the bound of (12) is also optimized in $\theta$.

---

[7]We do not comment more on how to tune $\rho$ with respect to distribution shift and refer to Remark 12. Furthermore, the correct pointwise distance used as a base for the Wasserstein distance (here the Euclidean distance) may be tuned for each problem for a large performance gain. Such a numerical study is out-of-scope for the present paper.

[8]the bound of (12) with a smoothed loss solved by entropic regularization

In Fig. 5, we report the bound values and the test risk (estimated from 1000 independent samples) for separate and joint training as a function of $\lambda$. We observe that the joint approach enables to have a smaller bound and risk than the separate one, for any value of $\lambda$. Hence, simultaneous training and conformal inference will lead to a better conformal inference performance at the cost of degrading the empirical performance.

In Fig. 6, we observe the models obtained by joint learning and conformal prediction can be very different than the one obtained separately by empirical risk minimization. Indeed, if $\lambda$ is small, it is difficult to obtain meaningful confidence guarantees and thus any model than can grab a few points will be optimal. On the over way around, if $\lambda$ is large, it is easy to be confident due to the large error margin and a simple tendency of the data is sufficient to obtain a good conformal prediction. In between, the situation is mixed as a sufficient number of points should be correctly predicted but whenever the prediction fails, it does not matter by how much and thus the conformal loss training offers some kind of regularization to the predictor.

Thus, we can see that training a model using the distributionally robust conformal loss does not lead to good test performance even though the conformal risk is improved. Nevertheless, this opens the door for the study of WDRO-based conformal risk as a regularization for the empirical risk minimization which could lead to good performance but with an improved conformal coverage.

**Remark 13** (On the interplay between coverage and performance). *There is a subtlety here in the interaction between conformal prediction (ie. risk control) which is at the core of this work and prediction performance of the base model which is not our main concern. The basic conformal prediction framework aims to quantify the uncertainty of a predictor (whatever its performance). Typically, for the regression case, if the predictor has poor performance for a given criteria and that the conformal interval is built with respect to this criteria, then the obtained prediction sets (ie. $\lambda$) will be large. Conversely, if we minimize the prediction size while jointly training a model (as we do in this section), then the obtained predictor should be somehow performing but has no reason to be optimal.[9]*

*When jointly training, the RCPS guarantees are theoretically preserved since the generalization part only depends on measure concentration. Intuitively, by doing so we should reduce the overfitting of the predictor even if we might "overfit" the prediction set, still this is mitigated by the distributionally robust optimization of the conformal loss. In addition, conformal prediction intervals can be obtained even when the models depend on the calibration data under some assumptions (see eg. Barber et al. (2023)).*

*Additional experiments exploring this setting can be found in Appendix A.*

## 5   Conclusion

In this paper, we discussed the design of risk-controlled prediction sets using Wasserstein distributionally robust optimization. We showed that, by replacing the model loss by the conformal loss, the WDRO problem exactly leads to the type of generalization guarantees needed in the considered conformal prediction approach. This demonstrates the potential links between these two separate methods which can foster future research. In addition, the obtained WDRO-based conformal prediction methods displayed promising performance in our numerical illustrations and benefit from two major advantages: i) they can seamlessly account for distribution shifts; and ii) they allow for simultaneous training and calibration on the same dataset which is a advantageous in data-poor situations.

**Acknowledgments**

The authors thank the action editor and the reviewers for their fruitful remarks. F. Iutzeler acknowledges the support of the AI Cluster ANITI (ANR-23-IACL-0002) and the ANR project MAD (ANR-24-CE23-1529). A.

---

[9]Note that we consider in this work the "conformal risk control" approach, which is a more general framework: as shown in Example 2.1, the size of the conformal interval is then generalized through the parameter $\lambda$. Both frameworks aim to get guarantees as Eq. (1) (ie. getting a prediction set for the new output at a given level $1 - \alpha$) which is then a guarantee in expectation. In particular, it implies that the level $1 - \alpha$ would not be achieved for some realizations of $(X_i, Y_i)_{i=1}^{n+1}$. The guarantees we are studying here (Eq. (9)) aim to ensure that the proportion of such events remains controlled at another level $1 - \delta$. This can be achieved by extrapolating concentration knowledge for the considered risk (Eq. (7)), as proposed by Bates et al. (2021).

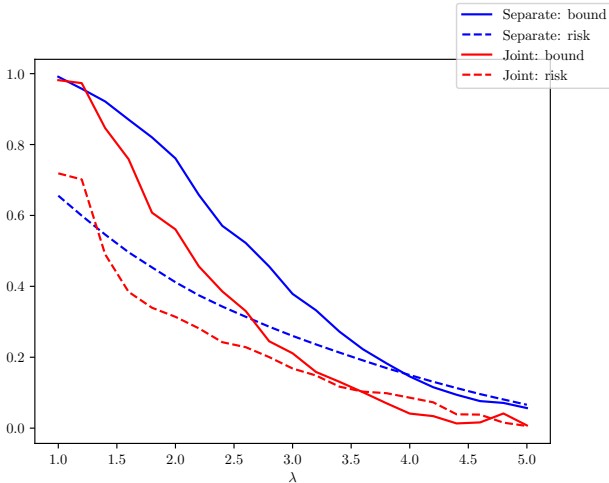

**Figure 5:** Conformal bounds and test risk (estimated from 1000 independent samples) for separate and joint training.

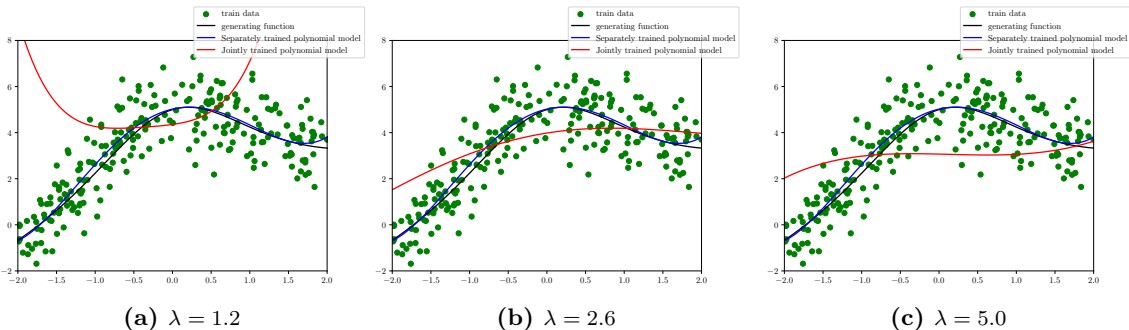

**(a)** $\lambda = 1.2$      **(b)** $\lambda = 2.6$      **(c)** $\lambda = 5.0$

**Figure 6:** Polynomial models learned for different values of $\lambda$ with separate and joint training.

Mazoyer has been supported by the project ROMEO (ANR-21-ASIA-0001) from the ASTRID joint program of the French National Research Agency (ANR) and the French Innovation and Defence Agency (AID).

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

# A   Extra numerical illustrations for joint training and conformal inference

In this section, we use the same setup as in Section 4.3 but change some of the parameters.

## A.1   Varying the number of samples

We take $n = 500, 1000, 2000$ samples and report the bounds and test risk in Fig. 7 and the obtain models in Fig. 8. We observe that neither the models nor the prediction change dramatically, indicating that the concentration observed is stable and hence the predictions. In particular, we do not particularly see overfitting behaviors.

## A.2   Changing the variance

We take a variance of $\sigma = .1$, .2, and .5 (instead of 1) in the Gaussian noise $U$ and report the bounds and test risk in Fig. 9 and the obtain models in Fig. 10. With a smaller variance, the prediction size needed to obtain a coverage guarantee diminishes as wee see in Fig. 9. Thus for $\lambda = 1.0$, we observe that the models are better adjusted than with unit variance and for the $\lambda = 2.6$ corresponding to an approximate coverage of 80%, the model better follows the curve compared to the unit variance case (which has a coverage of approximately 65%). Finally, for $\lambda = 5.0$, the size is so large that virtually any model will have a perfect coverage hence the observed behavior.

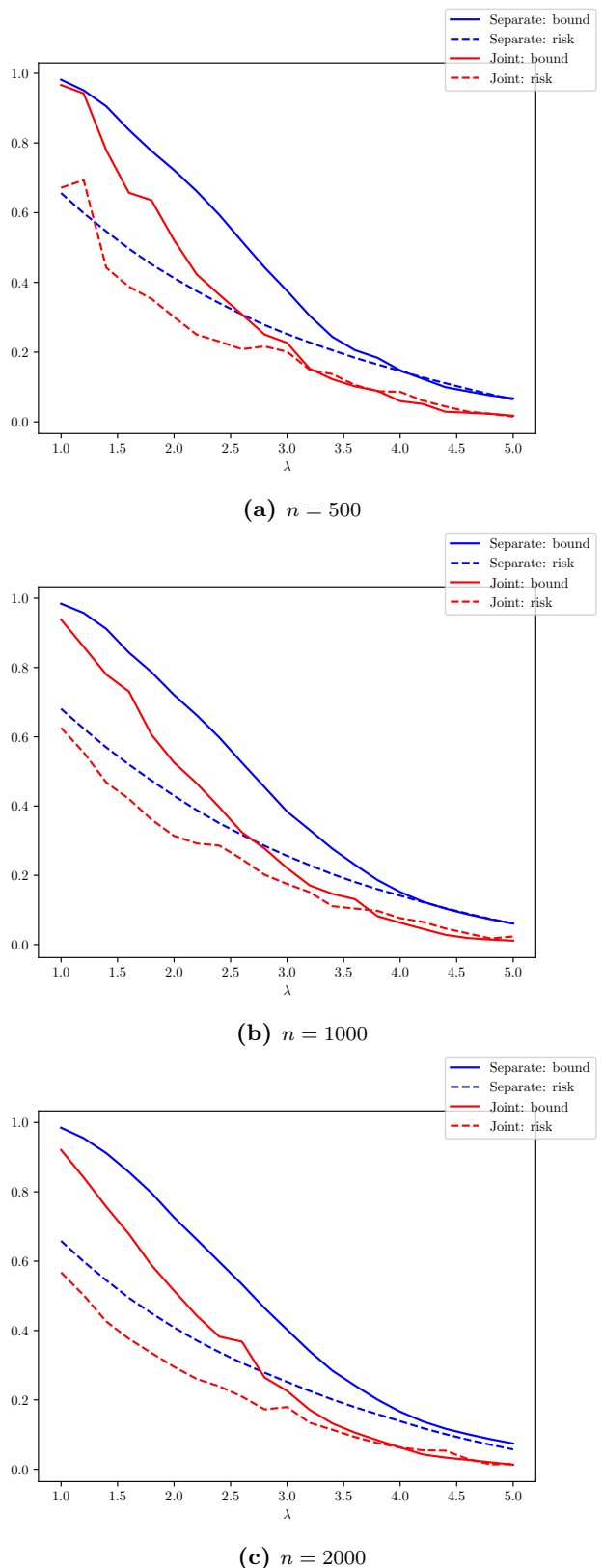

**(a)** $n = 500$

**(b)** $n = 1000$

**(c)** $n = 2000$

**Figure 7:** Conformal bounds and test risk for separate and joint training .

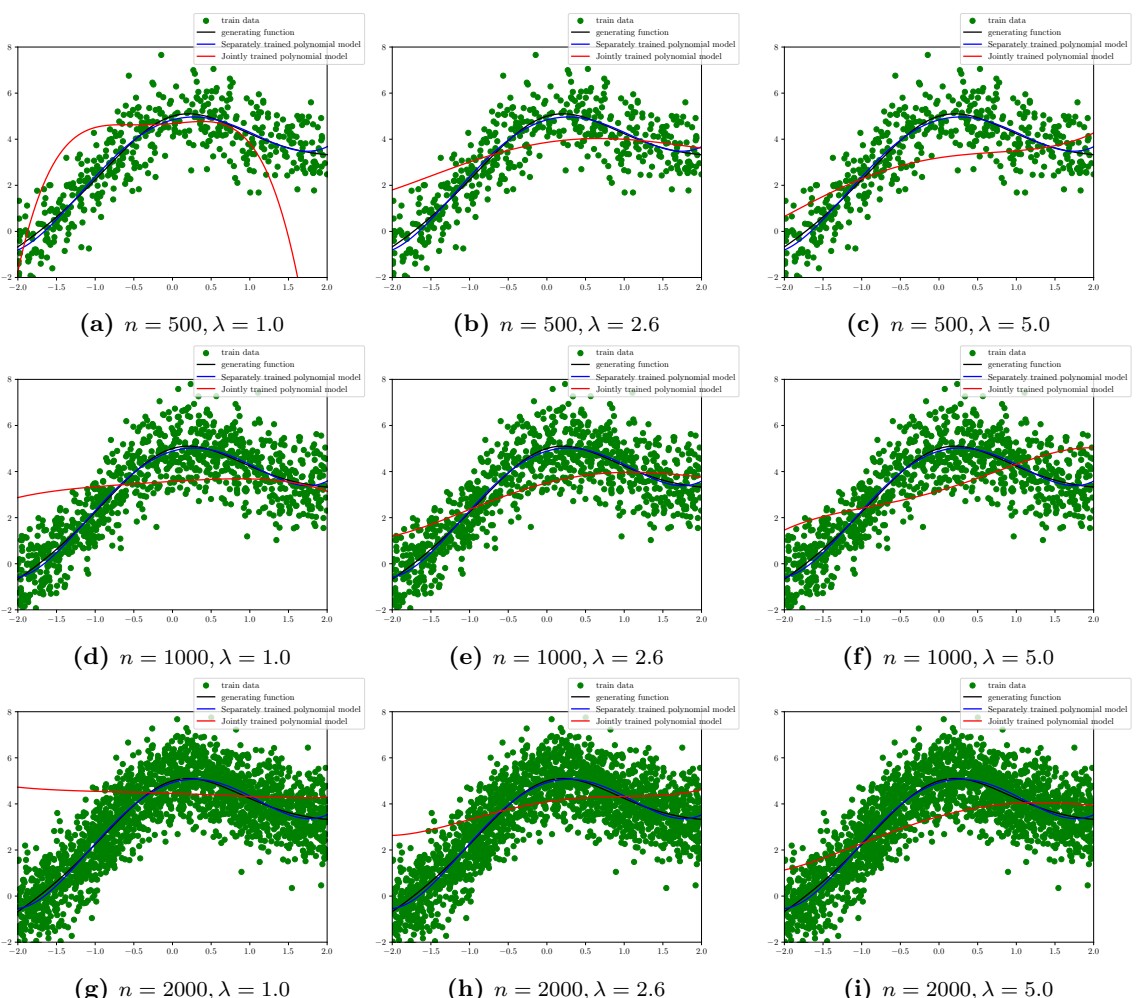

**Figure 8:** Polynomial models learned for different values of $\lambda$ with separate and joint training.

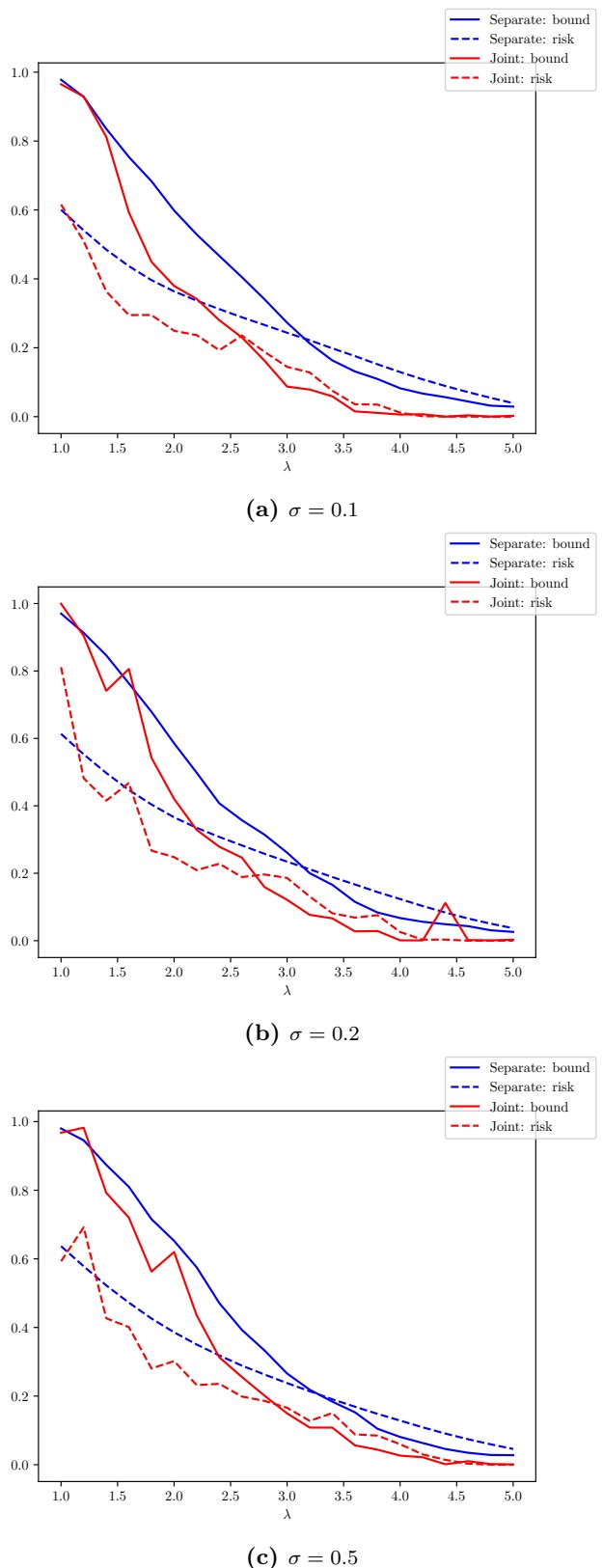

**(a)** $\sigma = 0.1$

**(b)** $\sigma = 0.2$

**(c)** $\sigma = 0.5$

**Figure 9:** Conformal bounds and test risk for separate and joint training .

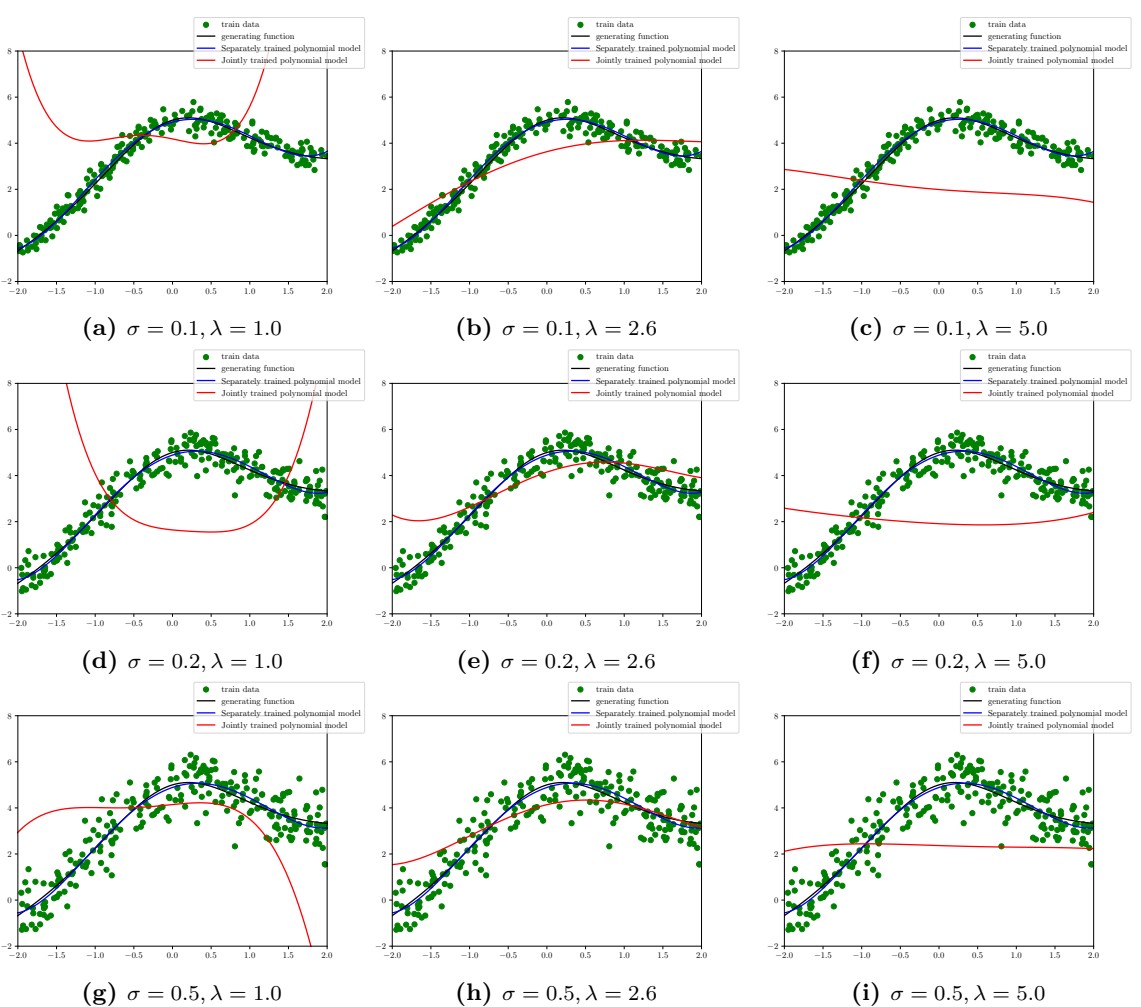

**Figure 10:** Polynomial models learned for different values of $\lambda$ with separate and joint training.

