# OpenReview forum: "Risk-controlling Prediction with Distributionally Robust Optimization"
_TMLR — Accepted by TMLR_

### Review · Reviewer_6jTx · 2025-06-17

**Summary Of Contributions:**

This paper proposes a new framework to construct risk-controlling prediction sets using Wasserstein Distributionally Robust Optimization (WDRO). Unlike classical conformal prediction methods, risk-controlling conformal prediction intervals can be learned jointly with the predictive model, under the WDRO formalism. The method claims to offer improved statistical guarantees and tighter prediction intervals. Theoretical results support the validity of the approach under certain assumptions, and experiments illustrate the benefits and trade-offs of the method.

**Audience:**

Yes

**Broader Impact Concerns:**

There are no additional ethical concerns identified that require discussion.

**Claims And Evidence:**

Yes

**Requested Changes:**

1. It is suggested to provide guidance on selecting the Wasserstein radius, including either heuristic or empirical strategies.
2. The degradation in predictive accuracy observed in Section 4.3 should be analyzed in more detail.
3. It is suggested to add a comparison of joint setups under varying sample sizes to examine if data reuse leads to overfitting.
4. It is suggested to provide more discussion or information on solving the regularized problem. For example, including a time comparison to baseline methods would be useful.

**Strengths And Weaknesses:**

**Strengths**
1. The paper establishes a formal connection between conformal risk control and distributionally robust optimization, providing uniform high-probability guarantees under Wasserstein shift.
2. The ability to simultaneously train the model and calibrate prediction intervals on the same dataset offers an improvement over classical conformal prediction.
3. The paper presents a tractable formulation for the WDRO objective and provides implementation strategies utilizing regularized approximations.

**Weaknesses**
1. The method’s validity depends on choosing the Wasserstein radius $\rho$, but the paper provides no strategy to select it.
2. In Section 4.3, models trained to minimize conformal loss achieve poor point prediction performance. This trade-off is acknowledged but not analyzed, limiting the method’s utility in tasks requiring accurate predictions.
3. Although the paper promotes joint training and calibration as a sample-efficient alternative, it does not assess the potential for overfitting when data is reused.

---

> ### Author Response · Authors · 2025-07-04
> **Response to Reviewer 6jTx**
>
> Thank you for you precise arguments, you will find below a point-by-point response to your comments. In turn, we will add several remarks clarifying the raised questions in the upcoming revision as well as additional experiments to clarify the computational cost and the scaling of the joint training procedure.
>
> ### Weaknesses
>
> 1. The choice of the radius is an important and difficult question in distributionally robust optimization. We can distinguish two approaches: (i) generic measure concentration (as used in [1] and Lemma 5), which gives $\rho=(log(C/\delta)/(cn))^{4/d}$ and thus suffers from the curse of dimensionality (but can be used directly to obtain statistical guarantees on WDRO estimators); or (ii) specific concentration on the WDRO (dual) objective which give $\rho = C'/n$ but are unfortunately elusive for the conformal objective functions considered in the paper (mainly due to the discontinuity, see [2]). In both cases, constants are not explicit. Nevertheless, as in classical WDRO we found that $\rho\propto 1/n$ was performing rather well all around and this was used in our experiments (see top of Sec. 4). This finding is in line with the ``practical rate'' for WDRO estimation as mentioned in Sec. 7.2.D of [1].
>
> 2. There is a subtlety here in the interaction between conformal prediction (ie. risk control) which is at the core of this work and prediction performance of the base model which is not our main concern here. The basic conformal prediction framework aims to quantify the uncertainty of a predictor (whatever its performance). Typically, for the regression case, if the predictor has poor performance for a given criteria and that the conformal interval is built with respect to this criteria, then the obtained prediction sets (ie. $\lambda$) will be large. Conversely, if we minimize the prediction size while jointly training a model (as in Sec. 4.3), then the obtained predictor should be somehow performing but has no reason to be optimal.
> Indeed, risk guarantees as eq. (1) are in expectation. In particular, it implies that the level $1-\alpha$ would not be achieved for some realizations of $(X_i,Y_i)_{i=1}^{n+1}$. The guarantees of eq. (9) aims to ensure that the proportion of such events remains controlled at another level $1-\delta$ which can be achieved by extrapolating concentration knowledge for the considered risk (eq. (7)), as proposed by [3].
>
> 3. When jointly training, the RCPS guarantees are theoretically preserved since the generalization part only depends on measure concentration. Intuitively, by doing so we should reduce the overfitting of the predictor even if we might "overfit" the prediction set, still this is mitigated by the distributionally robust optimization of the conformal loss. In addition, conformal prediction intervals can be obtained even when the models depend on the calibration data under some assumptions (see eg. [4]).
>
>
> Requested changes
>
> 1. As mentioned in Weakness 1, a practical rate $\propto 1/n$ is generally observed, we will emphasize this point (we discussed it slightly in the beginning of Section 4 and Remark 12). Furthermore, for a specific dataset, several approaches can be used to tune the radius such as out-of-sample performance (with houldout or cv) or reliability (see Sec. 7.2.B,C in [1]); these techniques can be alternatives to the standard $1/n$ choice. Nevertheless, in the case of an unknown distribution shift, these dedicated techniques could be too optimistic which is why we did not focus on this aspect.
>
> 2. See Weakness 2, we will add a remark clarifying this subtlety in the upcoming revision.
>
> 3. We will add the simulation for $n= 500,1000,2000$ in the appendix to illustrate the behavior of the prediction sets with $n$.
>
> 4. The regularized version takes about 10 times as much time as the original training (in joint training and prediction) in our experiments. The computational difficulty comes from the Monte-Carlo estimation of the expectation that replaces the supremum in (15). We will add the formula for the regularized objective to better discuss this point in Sec. 3.4.3 and include time-comparison with the other baselines in the experiments.
>
>
> [1] Mohajerin Esfahani, P. and Kuhn, D., 2018. Data-driven distributionally robust optimization using the Wasserstein metric: Performance guarantees and tractable reformulations. Mathematical Programming, 171(1), pp.115-166.
>
> [2] Azizian, W., Iutzeler, F. and Malick, J., 2023. Exact generalization guarantees for (regularized) wasserstein distributionally robust models. Advances in Neural Information Processing Systems, 36, pp.14584-14596.
>
> [3] S. Bates, A. Angelopoulos, L. Lei, J. Malik, and M. Jordan, 2021. Distribution-free, risk-controlling prediction sets. Journal of the ACM, 68(6), pp.1–34.
>
> [4] Barber, R.F., Candes, E.J., Ramdas, A. and Tibshirani, R.J., 2023. Conformal prediction beyond exchangeability. The Annals of Statistics, 51(2), pp.816-845.

---

> > ### Comment · Reviewer_6jTx · 2025-07-24
> >
> > Thank you to the authors for the detailed response. Most of my concerns have been acknowledged, and I appreciate the clarifications. I would still encourage the authors to provide more practical guidance on tuning the Wasserstein radius. The inclusion of a discussion on computational comparisons is appreciated. Overall, the proposed revisions adequately address the key concerns raised.

---

### Review · Reviewer_nLLr · 2025-07-13

**Summary Of Contributions:**

This paper combines the conformal risk-control framework with the distributionally robust optimization problem. Specifically, the method builds a Wasserstein ball around the empirical distribution and computes the worst-case expected loss over this set, leading to prediction intervals that remain reliable even when there is a moderate distribution shift between calibration and test data. Theoretical results, such as Theorem 1, establish that as long as the test distribution lies within this Wasserstein ball, one can control the predictive risk with high confidence. The extensive experiments show the superiority of WDRO-based upper bounds.

**Audience:**

Yes

**Claims And Evidence:**

Yes

**Requested Changes:**

1. In Figure 6, the jointly trained polynomial model exhibits a worse fit but achieves a lower bound value compared to the separately trained model. Could this be due to high noise levels? The authors are encouraged to provide an additional experiment using a standard noise variance of 0.1 instead of 1 to clarify this behavior.

2. Minor typos:

- Page 1: “reliablity” → “reliability”
- Page 14: Add a comma before “i.e.” in the "joint" section

**Strengths And Weaknesses:**

# Strengths

1. The paper is well-written and clearly structured. The key concepts are easy to follow and understand.
2. The proposed methodology is theoretically sound and rigorously developed. The authors provide a comprehensive theoretical analysis of WDRO-based risk-controlled prediction sets (RCPS). And then, the dual formulation in Section 3.4 transforms the infinite-dimensional robust risk optimization into a tractable finite-dimensional problem. In addition, the proposed simultaneous learning is also a good choice to furthe improve the efficiency of prediciton intervals.
3. The exmperical experiemts are extensive. The authors demonstare the uperioity of their method in the examples with and without distribution shift.

# Weaknesses

1. The authors should clarify the role of Section 3.4 in the overall methodology. Specifically, since the duality is claimed to be *“the computational scheme of WDRO”*, does the implementation in experiments directly solve the dual problem (e.g., Theorem 6) to estimate $\hat{R}^\rho_n(\lambda)$?

2. Some empirical results need to be further explained. For example, in the second subfigure of Figure 4, the average coverage under $\rho=0.5$ falls below $1-\alpha$, so the theoretical guarantee no longer applies.

---

> ### Author Response · Authors · 2025-07-24
> **Response to Reviewer nLLr**
>
> Thank you for your review. Please find below a point-by-point response to your comments.
>
> ## Weaknesses
>
> 1. The WDRO problem is a semi-infinite one in the sense that the optimization space is an infinite-dimensional set of measures with a finite number of constraints. In such problems, directly solving the primal problem (by relying on RKHS or Wavelet approximations) is often very cumbersome compared to solving the finite-dimensional dual problem. This is exactly the case here where the dual WDRO problem (15) is one-dimensional and thus much easier to solve. The only issue is that the objective is implicit which is why a regularized version as described in 3.4.3 is often solved. We have reworded some portions at the top of Section 3.4 and 4 to make this clearer.
>
> 2. Indeed, in the middle figure, $\rho=0.5$ does not attain the RCPS guarantee. This is due to the fact that the distribution shift was larger in magnitude than the WDRO radius making this situation unattainable for so small a radius. We have added a sentence detailing this after the figure.
>
> ## Requested changes
>
>
> 1. We have added an extra experiment in the appendix featuring different variance choices. For smaller variances, we have the same coverage with a reduced interval size and thus the prediction tends to be a bit better but still does not get close to the optimal predictor without conformal prediction. This is discussed in the new Remark 13.
>
> 2. Thank you, fixed.

---

### Review · Reviewer_uVd6 · 2025-07-16

**Summary Of Contributions:**

This paper draws a connection between conformal prediction and distributionally robust optimization (DRO) by pointing out that the "conformal set" over which a prediciton function is calibrated can be connected to the uncertainty set in DRO. Thus by using methods for conformalizing the risk of a function, we recover DRO w.r.t. the conformal set for free.

**Audience:**

No

**Claims And Evidence:**

Yes

**Requested Changes:**

See my notes above. I think this is likely a valuable connection to pursue. But merely proving the existence of choices of $\lambda$ and $\rho$ for conformal prediction without any sort of bound on how $\alpha$ or $\rho$ grow as a function of the shift does not feel sufficient, particular in cases where we assume e.g. that $\mathcal{X}\times\mathcal{Y}$ is bounded.

**Strengths And Weaknesses:**

While the connection between conformal prediction and DRO is a meaningful one to investigate, I unfortunately don't see much in this paper that I think is a major contribution.

The main results reason about the existence of choices of conformal parameters $\lambda$ which satisfy risk quantiles $\alpha$ over distributions within some distance defined by a function $\rho$. The paper makes no attempt to reason about what these values may be, hence, it is almost trivial by simply allowing $\alpha$ or $\rho$ to be arbitrarily large. In particular, I believe Assumption 3(a) *immediately* implies this result for Wasserstein-2 distance by simply letting $\rho$ be equal to the $\ell_2$ diameter of the space. Assumption 3(b) is similarly strong.

Essentially, by choosing a very weak notion of distribution shift (Wasserstein-2 distance) and never specifying any sort of limits on how conservative the predictor may need to be to satisfy conformal risk guarantees under that shift, the proven results reduce almost entirely to just the original observation: that conformal prediction can be "reparameterized" as DRO.

Furthermore, there seems to be little mathematical analysis in a paper focused purely on math. Most of the Lemmas are literal definitions or one-line proofs, and those that are not are taken (sometimes with no additional analysis) from the references. This includes Lemma 4 from Gao and Leywegt (2023), Lemma 5 from Fournier and Guillin (2015), and Theorem 10 from Yue et al (2022) and Gao and Leywegt (2023).

---

> ### Author Response · Authors · 2025-07-24
> **Response to Reviewer uVd6**
>
> We thank the reviewer for his remarks and acknowledging the interest of investigating the relation between conformal prediction and DRO.
>
>
> First of all, to be precise, the connection of interest here is with the risk-controlling prediction (RCPS) methodology (as mentioned in the title and Sec. 1.1), which is not exactly the split conformal prediction (SCI) viewpoint, as detailed in Sec. 1.2.
> The main point of the paper is to properly combine RCPS and WDRO methodologies:
> * RCPS aims to provide a theoretical control of a given parameterized risk/metric (in $\lambda$) on test data given (i) calibration data and (ii) theoretical concentration result. The RCPS guarantees the identification of a well chosen $\lambda$ according to the calibration data (ingredient (i)) to extrapolate a concentration result (ingredient (ii)) at a magnitude $1-\delta$ (set by the user) of the risk evaluated on the test data below a target risk level $\alpha$ (also set by the user). In particular, the methodology is actually agnostic to the predictor function or to the design of the validity intervals.
>
> * The main point of this work is then to rely on WDRO on the emprirical risk to obtain the associated concentration guarantees (by classical WDRO results), as ingredient (ii) in the RCPS methodology. Moreover, the numerical experiments of Sec. 4.1 tend to show that using that bound instead of ``naive'' ones (such that Hoeffding or Bernstein) offers a less conservative behavior, which is clearly desirable.
>
> * Finally, using WDRO for RCPS also comes with methodological advantages compared to usual methods. First, it enables to encompass distribution shifts by simply adjusting the robustness radius (as discussed in Sec. 3.3  and illustrated numerically in Sec. 4.2). Second, it enables to train the base predictor and select the validity interval using the same data (as discussed in Sec. 3.5 and illustrated numerically in Sec. 4.3), which is attractive in data scarse environments.
>
>
>
> *"The paper makes no attempt to reason about what these values may be, hence, it is almost trivial by simply allowing $\alpha$ or $\rho$ to be arbitrarily large. [...]"*
>
> As mentioned before, in the RCPS methodology, the target conformal risk $\alpha$ (intuitively, the proportion of points falling outside the validity interval in the test distribution) and magnitude $1-\delta$ (the probability that the proportion of points falling outside the validity interval in the test distribution falls below alpha over the training sample distribution) are set by the user (typically to $10\%$ and $90\%$). These parameters can indeed be fixed arbitrary large (for $\alpha$ or $\delta$) by the user, and the guarantee would still hold but the result would be completely uninformative.
>
> What we thus have to show in this paper is that a) for any radius $\rho$,  we can find a size $\lambda$ so that the WDRO conformal loss is smaller than $\alpha$ for all intervals greater than this size (this attainability and monotonicity is given by Lemma 7 and 8); and b) we can find $\rho$ so that  for any $\lambda$  the true risk is upper bounded by the WDRO risk with a prescribed probability (Lemma 5). This all relies only on mild properties on the test distribution and on Assumption 2 which is quite standard.
>
> In practice, $\alpha$ is thus fixed by the user, $\rho$ is given by the number of samples, confidence, and eventual shift. Then, the prediction size is chosen from the data and WDRO bound so that Eq. (8) is satisfied by grid-search (see Step B in Sec. 2.2). What we see numerically, is that using WDRO compared to classical concentration bounds often offers tighter prediction sizes at the price of a higher computational cost.
>
> Assumption 3 is simply here to avoid degenerate cases where the WDRO would be infinite for extreme values of $\rho$ and $\lambda$, which would prevent its numerical computation and make the search for $\lambda$ uneasy. Both points are classical in the WDRO literature and part i) is directly satisfied for the classical 0/1 conformal loss.
>
>
> *"Furthermore, there seems to be little mathematical analysis in a paper focused purely on math. Most of the Lemmas are literal definitions or one-line proofs, and those that are not are taken (sometimes with no additional analysis) from the references. "*
>
> That is correct: our work is more about putting a simple and valuable (comparing to other generic concentration bounds) connection between RCPS and WDRO than highlighting new major theoretical results. It corresponds then more to a methodological paper than a fully theoretical one. With this humble work, we have tried to emphasize, as the reviewer rightly pointed out, that the connection between DRO and RCPS (or possibly other conformal methods) could be of interest, and we believe this falls within the scope of TMLR.

---

> > ### Comment · Reviewer_uVd6 · 2025-08-01
> > **Response**
> >
> > Thanks for your response.
> >
> > Yes, I understand that $\alpha$ and $\delta$ are set by the user (I see my typo above, my mistake). My concern is that neither $\rho$ nor $\lambda$ are explicitly reasoned about, which means that the statement made in the main Theorem says almost nothing at all, and furthermore it seems almost directly implied by some of the assumptions. Maybe I wasn't totally clear in stating this concern before so I will try again and hopefully you will be able to respond and let me know whether or not my understanding is correct.
> >
> > > "What we thus have to show in this paper is that a) for any radius $\rho$, we can find a size $\lambda$ so that the WDRO conformal loss is smaller than $\alpha$ for all intervals greater than this size"
> >
> > This is obvious and immediate from Assumption 2. In fact it seems that Assumption 2 was explicitly designed in order to imply this statement. The problem is that this tells you nothing about how *large* this $\lambda$ must be as a function of $\rho, \alpha$. If $\lambda$ is too large this predictor is useless because it doesn't actually commit to anything (e.g., a classifier whose output set is always all of the classes).
> >
> > > "and b) we can find $\rho$ so that for any $\lambda$ the true risk is upper bounded by the WDRO risk with a prescribed probability (Lemma 5)."
> >
> > Suppose I set $\rho(n, \delta)$ equal to the radius of the set $\mathcal{X}\times\mathcal{Y}$ (which is finite by Assumption 3(a)). Then $\mathcal{\hat R_n^\rho}(\lambda)$ is just the maximum loss of $\mathcal{T}_\lambda$ among *all possible distributions*. The only way that this can be less than $\alpha$ is if we push $\hat\lambda_n$ to be extremely large, which is problematic as I pointed out above. So, I've now proven that there exists a $\rho(n, \delta)$ which satisfies the claim you make in the main theorem, and yet **I haven't said anything meaningful at all---because the predictor which satisfies the described property is completely useless.**
> >
> > Please let me know if you see an error in this line of reasoning.

---

> > > ### Author Response · Authors · 2025-08-02
> > > **Reply to the response**
> > >
> > > Thank you for your feedback.
> > >
> > > **About the prediction size $\lambda$.**
> > >
> > > In the RCPS methodology, the pertinent size is the $\hat{\lambda}_n$ which is the smallest value that guarantees that the upper bound function $\hat{\mathcal{R}}^+_n(\lambda)$ (which relies on the conformal set $X_i,Y_i$) gets below $\alpha$.
> > > The value of $\hat{\lambda}_n$ is usually not controlled and may be get large, we do agree that it is a limitation of the RCPS methodology (and maybe of conformal inference in general) but it is not really particular to our use of WDRO within this context.
> > >
> > > For instance, in the literature [1] introduced RAPS (Regularized Adaptive Prediction Sets) which adds a penalty to discourage large prediction sets in image classification in practice (to overcome this issue in practice); and very recently (ICML 2025), this limitation was studied in perspective of Bayesian quadrature [2].
> > >
> > > The difficulty of controlling $\lambda$ is also the reason why in numerical evaluations, displaying the prediction size (in regression) is very important. What we see experimentally it that the prediction sizes with WDRO-based bound tend to be a bit better that with usual bounds (Fig. 3) and furthermore that they do not suffer from "jumps" due to empirical quantiles (Fig. 2). In addition, our results of Sec. 3.2.3, although almost immediate I agree, guarantee that for any $\alpha$ we can find a suitable $\hat{\lambda}_n$ with our WDRO bound, which is not the case with Hoeffding or Bernstein bounds (as mentioned in the PS and seen after $\lambda=63$ in Fig. 2). To us, this means that it is interesting to consider the use of WDRO bounds to replace the classical bounds used in RCPS which is the message of the paper.
> > >
> > > We will include a summary of these arguments and references in the next revision.
> > >
> > > **About the WDRO radius $\rho$**
> > >
> > > You are right, if you set $\rho$ as you say, you obtain a guarantee over all possibles distributions on the space. Fixing $\alpha$, this means you seek to bound the (quantile of) the prediction error no matter the data distribution. This will inevitably lead to a very large $\hat{\lambda}_n$, and although meaningless, this would be the only acceptable answer to the RCPS problem.
> > >
> > > The way Lemma 5 is formulated is currently misleading and indeed you can find a obvious proof. In fact, what we say is that if $\rho$ is greater than the sum of i) the particular $\rho(n,\delta) \propto (\log(1/\delta)/n)^{4/d}$ (at the bottom of page 7) and ii) the eventual distribution shift, then the sough guarantee (13) holds. Thus, we have some knowledge about $\rho(n,\delta)$ (also discussed in Sec. 3.3). We will reformulate it in order to stress this fact in the next revision.
> > >
> > > **References**
> > >
> > > [1] Angelopoulos, A., Bates, S., Malik, J. and Jordan, M.I., 2020. Uncertainty sets for image classifiers using conformal prediction. arXiv preprint arXiv:2009.14193.
> > >
> > > [2] Snell, J.C. and Griffiths, T.L., 2025. Conformal prediction as bayesian quadrature. arXiv preprint arXiv:2502.13228. ICML 2025.
> > >
> > > **PS: About the value of the prediction size in a simple example**
> > >
> > >
> > > In more details, if we place ourselves in the setting of Example 2.1 and use the Hoeffding bound ie. $\hat{\mathcal{R}}^+_n(\lambda) = $
> > >
> > >  $ \frac{1}{n} \sum_{i=1}^n 1_{|y_i-f_{\theta}(x_i)|>\lambda} $ $  + \sqrt{\log(1/\delta)/2n}$, then  the prediction size   is the $1-\alpha+\sqrt{\log(1/\delta)/2n}$ quantile of the absolute values of the errors (the $|y_i-f_{\theta}(x_i)|$) on the conformal set (note that with Hoeffding we need to have $\alpha>\sqrt{\log(1/\delta)/2n}$). The value of this quantile depends heavily i) on the predictor (which is taken fixed in the RCPS framework) and ii) on the data distribution.
> > > This means that i) without any noise, if $Y=1$ for all $X$ but the predictor is $f(X)=0$, then the prediction size must be $1$; ii) with a Gaussian model $Y=f(X)+N$ with $N$ a standard Gaussian noise, using $f$ as our predictor means that $\hat{\lambda}_n$ will be (asymptotically) normally distributed and it is thus difficult to say much else than its mean and variance.
> > >
> > > From these two reasons, it is hard to say much about the value of $\hat{\lambda}_n$ in RCPS in general (in fact, as I understand it, it goes a bit against the method's philosophy of being model agnostic). This is why an Assumption such as Assumption 2 is very common in conformal prediction: it is there to ensure that the prediction sets are designed so that by increasing the size we can get to (almost) any level of confidence.

---

### Author Response · Authors · 2025-07-24
**Revised version of the manuscript**

We thank the reviewers for their constructive comments. Based on their remarks, we have revised the paper in the following way:
* We have significantly expanded Remark 12 concerning the theoretical and practical choices for the WDRO radius $\rho$.
* We have added a remark in Section 4.3 (Remark 13) in order to clarify the interplay between coverage and performance when joint learning and performing conformal prediction on an estimator.
* We enhaced Section 3.4.3 by adding the formula for the regularized dual objective and discussing its computational cost.
* We reworded and slightly enhance the top of Section 3.4 in order to emphasize the importance of duality in WDRO. We also changed slightly the top of Section 4 in order to emphasize that the objectives numerically implemented are dual-based (and we now point directly to the refularized dual formula added in Section 3.4.3).
* We have added a discussion on Figure 4 at then end of Sec. 4.2 to better explain why the RCPS guarantees are not attained after shift for the smaller values of $\rho$.

---

> ### Author Response · Authors · 2025-08-21
> **Second revision**
>
> We thank the reviewers and the action editor for the fruitful interactions.
>
> Based on the discussion and the recommendation of the AE, we made the following modifications in the camera ready version of the paper:
> * We added a link to the GitHub of the experiments' code
> * We added some discussion about the provenance of mathematical results of Lemma 4 and 5, and we emphasized this point just after the main result (Theorem 9)
> * We stressed the technical and methodological originality of the WDRO approach in RCPS, notably at the end of Sec 1.1 where an outline has been added, highlighting the messages of the paper
> * We reworded part of the statement of Lemma 5 following a comment by Rev. uVd6 in order to clarify its role

---

### Decision · Action_Editor_tVYu · 2025-08-04

**Recommendation:** Accept with minor revision

**Additional Comments:**

After reading the authors’ exchange with the reviewers, I believe the paper needs another round of revisions before it can be accepted.

I share Reviewer uVd6’s concern that several results appear to have been compiled from earlier work and presented as new.

I encourage the authors to revisit the manuscript with a more critical eye. In particular, they should (i) state clearly the provenance of each mathematical result and (ii) focus on---and, where possible, further develop---new applications of their formulation.

**Audience:**

Yes

**Audience Explanation:**

DRO and predictive modelling are active ML fields, of interest to TMLR readers.

**Claims And Evidence:**

Yes

**Claims Explanation:**

The claims in the submission meet the TMLR threshold.